# Low-Precision Streaming PCA

**Sanjoy Dasgupta**
University of California San Diego
sadasgupta@ucsd.edu

**Syamantak Kumar**
University of Texas at Austin
syamantak@utexas.edu

**Shourya Pandey**
University of Texas at Austin
shouryap@utexas.edu

**Purnamrita Sarkar**
University of Texas at Austin
purna.sarkar@utexas.edu

## Abstract

Low-precision Streaming PCA estimates the top principal component in a streaming setting under limited precision. We establish an information-theoretic lower bound on the quantization resolution required to achieve a target accuracy for the leading eigenvector. We study Oja's algorithm for streaming PCA under linear and nonlinear stochastic quantization. The quantized variants use unbiased stochastic quantization of the weight vector and the updates. Under mild moment and spectral-gap assumptions on the data distribution, we show that a batched version achieves the lower bound up to logarithmic factors under both schemes. This leads to a nearly *dimension-free* quantization error in the nonlinear quantization setting. Empirical evaluations on synthetic streams validate our theoretical findings and demonstrate that our low-precision methods closely track the performance of standard Oja's algorithm.

## 1 Introduction

Quantization (or discretization) is the mapping of a continuous set of values to a small, finite set of outputs close to the original values; standard methods for quantization include rounding and truncation. The current popularity of training large-scale Machine Learning models has brought a renewed focus on quantization, though its origins go back to the 1800s. Some early examples include least-squares methods applied to large-scale data analysis in the early nineteenth century [Sti86]. In 1867, discretization was introduced for the approximate calculation of integrals [Rie67], and the effects of rounding errors in integration were examined in 1897 [She97]. For an excellent survey and history of quantization, see [GKD+22].

In the context of efficient model training, it is natural to ask the following: does training a model require the full precision of 32- or 64-bit representation, or is it possible to achieve comparable performance using significantly fewer bits? Mixed-precision training (using 16-bit floats with 32-bit accumulators) is now standard on GPUs and TPUs, yielding $1.5\times$ to $3\times$ speedups with negligible accuracy loss on large transformers and CNNs [MNA+18]. Binary Neural Networks (BNNs), which constrain weights and activations to $\pm 1$, can achieve up to $32\times$ memory compression and replace multiplications with bitwise operations. This has been shown to approach nearly full-precision ImageNet accuracy with careful training [HCS+16].

Theoretical analysis of the effect of low-precision computation on optimization problems has received significant attention [LD19, AGL+17, SZOR15, SLZ+18, LDX+17, ZLK+17]. Complementary strategies leverage stochastic rounding to mitigate quantization bias during LLM training. Ozkara *et al.* [OYP25] present theoretical analyses of implicit regularization and convergence properties of Adam when using BF16 with stochastic rounding, demonstrating up to $1.5\times$ throughput gains and 30% memory reduction over standard mixed precision [OYP25].

39th Conference on Neural Information Processing Systems (NeurIPS 2025).

Consider the set of values that can be exactly represented in the quantization scheme, which we call the *quantization grid*. For example, fixed-point arithmetic [Yat09] uses linear quantization (LQ), where the quantization grid consists of points spaced uniformly at a distance $\delta$ (also denoted by *quanta*). [LDX$^+$17] analyze Stochastic Gradient Descent (SGD)-based optimization algorithms for LQ, and [SYK21] perform Learned Image Compression (LIC) under 8-bit fixed-point arithmetic. Nonlinear quantization (NLQ) grids with logarithmic spacing are also widely used [KWW$^+$17, NTSW$^+$22, XLY$^+$24, YIY21, ZMK22, ZWG$^+$23] in low-precision training.

To illustrate the importance of the quantization scheme, consider the example of rounding, where each input is mapped to the value in the quantization grid closest to it. The following toy iterative optimization algorithm demonstrates that rounding can cause the solution to remain stuck at the initial vector. Consider the update scheme $\mathbf{w}_t = \mathbf{w}_{t-1} + \eta \mathbf{g}_t$, followed by rounding each coordinate of $\mathbf{w}_t$. Here $\eta$ is the learning rate and $\mathbf{g}_t$ is the gradient evaluated at time $t$. Suppose $\max_i \|\mathbf{g}_t(i)\| \leqslant 1$. Assume that $\mathbf{w}_0$ is quantized using the LQ scheme and that $\eta < \delta/2$. For any coordinate $i$, we have $|\mathbf{w}_1(i) - \mathbf{w}_0(i)| = \eta \cdot |\mathbf{g}_t(i)| \leqslant \eta$. Since $\eta < \delta/2$, after rounding, $\mathbf{w}_1(i)$ is mapped back to the original quantized value $\mathbf{w}_0(i)$, i.e., $\mathbf{w}_1 = \mathbf{w}_0$. As a result, the algorithm fails to make progress. We address this issue by using *stochastic rounding*. In this approach, each value is randomly mapped to one of the closest two quanta with the probabilities chosen such that the quantized value is unbiased.

**Principal Component Analysis.** PCA [Pea01, Zie03] is a dimension-reduction technique that extracts the directions of largest variance from the data. Suppose we observe $n$ independent samples $\mathbf{X}_i \in \mathbb{R}^d$ from a zero-mean distribution with covariance $\mathbf{\Sigma}$. PCA seeks a unit vector $\mathbf{v}_1$ that maximizes variance, which is any eigenvector of $\mathbf{\Sigma}$ associated with its largest eigenvalue $\lambda_1$. Under mild tail conditions on the $\mathbf{X}_i$, the top eigenvector $\hat{\mathbf{v}}$ of the sample covariance $\frac{1}{n}\sum_{i=1}^n \mathbf{X}_i\mathbf{X}_i^\top$ is a nearly rate-optimal estimator of the true principal direction $\mathbf{v}_1$ [Wed72, JJK$^+$16, Ver10].

Despite its statistical appeal, constructing the covariance matrix itself takes $\Omega(nd^2)$ time and $\Omega(d^2)$ space, which is prohibitive for large $d$ and $n$. A popular remedy is Oja's algorithm [Oja82], a *single-pass streaming algorithm* inspired by Hebbian learning [Heb49]. Starting from a (random) unit vector $\mathbf{u}_0$, for each incoming datum $\mathbf{X}_i$ the algorithm performs the update

$$\mathbf{u}_i \leftarrow \mathbf{u}_{i-1} + \eta\,\mathbf{X}_i\big(\mathbf{X}_i^\top \mathbf{u}_{i-1}\big), \qquad \mathbf{u}_i \leftarrow \mathbf{u}_i/\|\mathbf{u}_i\|. \tag{1}$$

Here, $\eta > 0$ is the learning rate which may vary across iterations. The batched version of Oja's method partitions the data into $b$ batches $B_1, \ldots B_b$ of size $n/b$ each and replaces the above update with the averages of the gradients within a batch:

$$\mathbf{u}_i \leftarrow \mathbf{u}_{i-1} + \eta\frac{\sum_{j\in B_i}\mathbf{X}_j\big(\mathbf{X}_j^\top \mathbf{u}_{i-1}\big)}{n/b}, \qquad \mathbf{u}_i \leftarrow \mathbf{u}_i/\|\mathbf{u}_i\|. \tag{2}$$

The entire procedure completes in $O(nd)$ time and uses $O(d)$ space. The scalability and simplicity of Oja's algorithm have motivated extensive analysis across statistics, optimization, and theoretical computer science [JJK$^+$16, AZL17, CYWZ18, YHW18, HW19, MP22, Mon22, KS24b, KS24a, JKL$^+$24, KPS25]. These works establish precise convergence rates, error bounds under various noise models, and extensions to sparse or dependent-data settings. When operating with $\beta$ bits, the overall complexity for streaming PCA (and that of the batched variant) grows polynomially with $\beta$ (for fixed $n, d$); Table 1 gives evidence towards this fact.

| | 64 bits | 16 bits |
|---|---|---|
| **Runtime (s)** | $0.0274 \pm 0.00136$ | $0.000398 \pm 0.0000235$ |

Table 1: Benchmarking runtimes[1] for the experiment described in Appendix F.1

**Our Contributions.**

1. We present a general theorem for streaming PCA with iterates that are composed of independent data (as in standard Oja's algorithm) and a noise vector that is mean zero, conditioned on the filtration up until now, which may be of *independent interest*.

2. We obtain new *lower bounds* for estimating the principal eigenvector under both quantization schemes. The quantization error depends linearly in the dimension $d$ for the linear scheme and dimension-independent (up to logarithmic factors) for the non-linear scheme.

---

[1]The experiments were conducted by representing the data and intermediate variables in double precision (64 bits) and half precision (16 bits) datatypes.

3. Our batched version of Oja's algorithm matches the lower bounds under both quantization schemes. The quantization error of the batched version with logarithmic quantization is *nearly dimension-free*. We also provide a procedure to make the failure probability of the algorithm arbitrarily small.

Section 2 introduces the problem setup and defines the linear and logarithmic quantization schemes. Section 3 presents the main results, including lower and upper bounds for Oja's algorithm with and without batching for both quantization schemes. Section 4 provides proof sketches, Section 5 reports experimental results, and Section 6 concludes the paper.

## 2  Problem Setup and Preliminaries

We use $[n]$ to denote $\{i \in \mathbb{N} \mid i \leqslant n\}$. Scalars are denoted by regular letters, while vectors and matrices are represented by boldface letters. $\mathbf{I} \in \mathbb{R}^{d \times d}$ represents the $d$-dimensional identity matrix. $\|.\|$ denotes the $\ell_2$ euclidean norm for vectors and $\|.\|_{\mathrm{op}}$ denotes the operator norm for matrices. For $a, b \in \mathbb{R}$, we write $a \lesssim b$ if and only if there exists an absolute constant $C > 0$ such that $a \leqslant Cb$. $\tilde{O}, \tilde{\Omega}$ represent order notations that hide logarithmic factors. $\mathbb{S}^{d-1}$ is the set of unit vectors in $\mathbb{R}^d$.

We operate under the following assumption on the data distribution.

**Assumption 1.** $\{\mathbf{X}_i\}_{i \in [n]}$ *are mean-zero* iid *vectors in* $\mathbb{R}^d$ *drawn from distribution* $\mathcal{D}$ *supported on the unit ball. Let* $\boldsymbol{\Sigma} := \mathbb{E}_{\mathbf{X} \sim \mathcal{D}} \left[ \mathbf{X}\mathbf{X}^\top \right]$ *denote the data covariance, with eigenvalues* $\lambda_1 > \lambda_2, \cdots, \lambda_d$ *and corresponding eigenvectors* $\mathbf{v}_1, \mathbf{v}_2, \cdots \mathbf{v}_d$. *We assume* $\exists \mathcal{V}, \mathcal{M} > 0$ *such that*

$$\mathbb{E}_{\mathbf{X} \sim \mathcal{D}}[\|\mathbf{X}\mathbf{X}^\top - \boldsymbol{\Sigma}\|^2] \leqslant \mathcal{V} \text{ and } \left\| \mathbf{X}\mathbf{X}^\top - \boldsymbol{\Sigma} \right\|_2 \leqslant \mathcal{M} \text{ almost surely for } \mathbf{X} \sim \mathcal{D}.$$

Assumption 1 enforces standard moment bounds used to analyze PCA in the stochastic setting. Similar assumptions are also used in [HP14, SRO15, Sha16a, Sha16b, JJK$^+$16, AZL17, BDWY16, XHDS$^+$18] to derive near-optimal sample complexity bounds for Oja's rule. We assume a bounded range for ease of analysis, and it can be generalized to subgaussian data (see [LSW21, KS24a, Lia21]).

The misalignment between the estimated top eigenvector $\mathbf{u}$ and the true eigenvector $\mathbf{u}_1$ is measured using the *principal angle* between the two vectors. The *sin-squared error* between any two non-zero vectors $\mathbf{u}, \mathbf{v}$ is defined as $\sin^2(\mathbf{u}, \mathbf{v}) = 1 - \frac{(\mathbf{u}^\top \mathbf{v})^2}{\|\mathbf{u}\|^2 \|\mathbf{v}\|^2}$.

### 2.1  Quantization Schemes and Rounding

**Linear quantization**: Let $\delta > 0$, and let $\beta > 0$ be the number of bits used by the low-precision model to represent numbers. A linear quantization scheme uniformly spaces on the real line. Define

$$\mathcal{Q}_L(\delta, \beta) := \left\{ -\delta 2^{\beta-1}, -\delta(2^{\beta-1} - 1), \ldots, -\delta, 0, \delta, \ldots, \delta(2^{\beta-1} - 1) \right\}. \tag{3}$$

We call $\delta$ the *quantization gap* for the *quantization grid* $\mathcal{Q}_L$.

**Logarithmic (non-linear) quantization**: The error resulting from rounding an element $x$ in the range $\left[ -\delta 2^{\beta-1}, \delta(2^{\beta-1} - 1) \right]$ using the linear quantization scheme is an additive $\delta$. Here, we present a well-known non-linear quantization scheme where the error scales with the quantized value.

The quantization grid $\mathcal{Q}_{NL}$ in the *logarithmic quantization* scheme with parameters $\zeta$ and $\delta_0$ is defined as follows: Let $q_0 = 0$ and $q_{i+1} = (1 + \zeta)q_i + \delta_0 \ \forall i \in \mathbb{N}$. Then,

$$\mathcal{Q}_{NL}(\zeta, \delta_0, \beta) := \left\{ -q_N, -q_{N-1}, \ldots, -q_1, q_0, q_1, \ldots, q_{N-1} \right\}, \tag{4}$$

where $N = 2^{\beta-1}$. Henceforth, non-linear quantization refers to logarithmic quantization.

These two quantization schemes are widely used in practice [YIY21, DSLZ$^+$18, LDS19, DMM$^+$18]. Our analysis of the logarithmic scheme lifts to floating-point quantization commonly used in low-precision computing. The Floating Point Quantization (FPQ) is a widely adopted variation on the Logarithmic quantization scheme, where adjacent values in the quantization grid are multiplicatively close. FPQ and other logarithmic schemes are used in most modern programming languages such as C++, Python, and MATLAB, and broadly standardized (IEEE 754 floating-point standard [Kah96]).

Another quantization scheme for low-precision training is the power-of-two quantization [PRSS$^+$22], which rounds to the nearest power of two. All these schemes are similar in principle to our scheme; Lemma A.9 in the appendix establishes a relationship between the distance of a vector from its

quantization under NLQ. This Lemma applies to FPQ and to most other logarithmic quantization schemes. Our proofs can be modified to work with any such scheme.

**Stochastic Rounding.** A natural quantization scheme is to round $x$ to any of the closest values in the quantization grid. We can randomize to ensure that the expectation of the quantized number is equal to $x$. For this, we use a stochastic rounding scheme. For any $x$ within the range of the quantization grid $\mathcal{Q}$, suppose $u$ and $\ell$ are adjacent values in $\mathcal{Q}$ such that $\ell \leqslant x < u$. Define

$$\mathsf{Q}(x, \mathcal{Q}) = \begin{cases} \ell & \text{with probability } 1 - p(x) \\ u & \text{with probability } p(x) \end{cases}, \tag{5}$$

where $p(x) := (x - \ell)/(u - \ell)$. This choice of probability ensures

$$\mathbb{E}\left[\mathsf{Q}(x, \mathcal{Q}_{NL})|x\right] = x, \ |\mathsf{Q}(x, \mathcal{Q}_{NL}) - x| \leqslant u - \ell, \ \text{Var}(\mathsf{Q}(x, \mathcal{Q}_{NL})|x) \leqslant (u - \ell)^2/4. \tag{6}$$

## 3 Main Results

### 3.1 Lower Bounds

In this section, we establish worst-case lower bounds for the quantized PCA for both linear and logarithmic quantization schemes under the mild assumption that the quantized vectors under consideration have bounded norm. This assumption is reasonable because (i) gradient-based algorithms and other typical algorithms for PCA are usually self-normalizing, ensuring that the norms of the iterates are controlled, and (ii) the quantized vectors are close to the true vectors in norm.

**Lemma 1.** *[Lower bound for linear quantization] Let $d > 1$ and $\delta > 0$ such that $\delta^2 d \leqslant 0.5$. Let $\mathcal{V}_L$ denote the set of non-zero quantized vectors $\mathbf{w} \in \mathbb{R}^d$ using the linear quantization scheme* (3) *such that $\|\mathbf{w}\| \in [1/2, 2]$. Then, $\sup_{\mathbf{v}_1 \in \mathbb{S}^{d-1}} \inf_{\mathbf{w} \in \mathcal{V}_L} \sin^2(\mathbf{w}, \mathbf{v}_1) = \Omega(\delta^2 d)$.*

**Lemma 2.** *[Lower bound for logarithmic quantization] Let $d > 1$ and $\delta_0, \zeta > 0$ such that $\zeta < 0.1$ and $\delta_0^2 d < 0.5$. Let $\mathcal{V}_{NL}$ be the set of non-zero quantized vectors $\mathbf{w} \in \mathbb{R}^d$ using the logarithmic scheme* (4) *such that $\|\mathbf{w}\| \in [1/2, 2]$. Then, $\sup_{\mathbf{v}_1 \in \mathbb{S}^{d-1}} \inf_{\mathbf{w} \in \mathcal{V}_{NL}} \sin^2(\mathbf{w}, \mathbf{v}_1) = \Omega(\zeta^2 + \delta_0^2 d)$.*

At first glance, the results of Lemmas 1 and 2 may appear similar. However, the parameter $\delta_0$ is substantially smaller than $\delta$. In Section 3.4, we select optimal values for $\delta$, $\delta_0$, and $\zeta$ given a fixed bit budget $\beta$ for the low-precision model and show that $\delta^2 d = \Theta(d4^{-\beta})$ while $\zeta^2 + \delta_0^2 d = \tilde{\Theta}(4^{-\beta})$ where the tilde hides a $\log^2 d$ factor. Hence, the lower bound for the logarithmic quantization scheme is *nearly independent* of the dimension. The proofs of the lower bounds are deferred to Appendix B.

### 3.2 Quantized Batched Oja's Algorithm

In this section, we present an algorithm that uses stochastic quantization for the batch version of Oja's algorithm (see Eq 2). We start by computing the quantized version $\mathbf{w}_i$ of the normalized vector $\mathbf{u}_{i-1}$ from the last step. Then, we quantize each $\mathbf{X}_j(\mathbf{X}_j^T \mathbf{w}_{i-1})$ and compute the average of the quantized gradient updates. This average gradient is quantized again and added to $\mathbf{w}_i$.

---

**Algorithm 1** Quantized Oja's Algorithm with Batches

---

**Require:** Data $\{\mathbf{X}_i\}_{i \in [n]}$, quantization grid $\mathcal{Q}$, learning rate $\eta$, number of batches $b$

1: Initialize $\mathbf{u}_0$ with a unit vector picked uniformly from $\mathbb{S}^{d-1}$.
2: $B_i \leftarrow \left\{(i-1)\frac{n}{b} + 1, \ (i-1)\frac{n}{b} + 2, \ \ldots, \ i\frac{n}{b}\right\}$
3: **for** $i = 1$ to $b$ **do**
4:      $\mathbf{w}_i \leftarrow \mathsf{Q}(\mathbf{u}_{i-1}, \mathcal{Q})$                              $\triangleright \boldsymbol{\xi}_{1,i} := \mathsf{Q}(\mathbf{u}_{i-1}, \mathcal{Q}) - \mathbf{u}_{i-1}$
5:      $\mathbf{z}_i \leftarrow \frac{\sum_{j \in B_i} \mathsf{Q}(\mathbf{X}_j(\mathbf{X}_j^T \mathbf{w}_i), \mathcal{Q})}{n/b}$         $\triangleright \boldsymbol{\xi}_{a,j,i} := \mathsf{Q}(\mathbf{X}_j(\mathbf{X}_j^T \mathbf{w}_i), \mathcal{Q}) - \mathbf{X}_j(\mathbf{X}_j^T \mathbf{w}_i)$
6:      $\mathbf{y}_i \leftarrow \mathsf{Q}(\eta \frac{\sum_{j \in B_i} \mathsf{Q}(\mathbf{X}_j(\mathbf{X}_j^T \mathbf{w}_i), \mathcal{Q})}{n/b}, \mathcal{Q})$          $\triangleright \boldsymbol{\xi}_{a,i} := \frac{\sum_{j \in B_i} \boldsymbol{\xi}_{a,j,i}}{n/b}$
7:      $\mathbf{u}_i \leftarrow \mathbf{w}_i + \mathbf{y}_j$                                   $\triangleright \boldsymbol{\xi}_{2,i} := \mathsf{Q}(\mathbf{y}_i, \mathcal{Q}) - \mathbf{y}_i$
8:      $\mathbf{u}_i \leftarrow \frac{\mathbf{u}_i}{\|\mathbf{u}_i\|}$
9: $\mathbf{w} \leftarrow \mathsf{Q}(\mathbf{u}_b, \mathcal{Q})$
10: **return** $\mathbf{w}$

---

The final vector that results from the batched Oja's rule (Eq 2) without quantization is

$$\mathbf{u}_{\text{unquantized}} = \frac{(\mathbf{I} + \eta \mathbf{D}_b) \dots (\mathbf{I} + \eta \mathbf{D}_2)(\mathbf{I} + \eta \mathbf{D}_1)\mathbf{u}_0}{\|(\mathbf{I} + \eta \mathbf{D}_b) \dots (\mathbf{I} + \eta \mathbf{D}_2)(\mathbf{I} + \eta \mathbf{D}_1)\mathbf{u}_0\|} = \frac{\prod_{i=b}^{1}(\mathbf{I} + \eta \mathbf{D}_i)\mathbf{u}_0}{\left\|\prod_{i=b}^{1}(\mathbf{I} + \eta \mathbf{D}_i)\mathbf{u}_0\right\|},$$

where $\mathbf{D}_i = \sum_{j \in B_i} \mathbf{X}_j \mathbf{X}_j^T / (n/b)$ is the empirical covariance matrix of the $i^{\text{th}}$ batch. Since $\mathbf{X}_i$ are IID and the batches are disjoint, $\mathbf{D}_i$ are also IID. The key observation for Algorithm 1 is that even with the quantization, the vector $\mathbf{u}_b$ can be written as

$$\mathbf{u}_b = \frac{\prod_{i=b}^{1}(\mathbf{I} + \eta \mathbf{D}_i + \boldsymbol{\Xi}_i)\mathbf{u}_0}{\| \prod_{i=b}^{1}(\mathbf{I} + \eta \mathbf{D}_i + \boldsymbol{\Xi}_i)\mathbf{u}_0\|}. \tag{7}$$

Each $\boldsymbol{\Xi}_i$ is a rank-one matrix resulting from the stochastic quantization. Conditioned on an appropriately chosen filtration $\sigma(\mathbf{X}_1, \dots, \mathbf{X}_i, \mathbf{u}_0, \dots, \mathbf{u}_{i-1})$, $\boldsymbol{\Xi}_i$ is mean zero; Algorithm 1 defines quantization variables $\boldsymbol{\xi}_{1,i}, \boldsymbol{\xi}_{a,i}$, and $\boldsymbol{\xi}_{2,i}$ for all $i \in [b]$. The rank one noise $\boldsymbol{\Xi}_i$ is $\boldsymbol{\Xi}_i := (\eta \boldsymbol{\xi}_{a,i} + \boldsymbol{\xi}_{2,i} + (\mathbf{I} + \eta \mathbf{D}_i)\boldsymbol{\xi}_{1,i})\mathbf{u}_{i-1}^T$. Since the stochastic updates are conditionally unbiased (equation (6)),

$$\mathbb{E}[\boldsymbol{\xi}_{1,i}|\mathbf{D}_1, \dots, \mathbf{D}_i, \mathbf{w}_0, \dots, \mathbf{w}_{i-1}] = 0.$$

Similarly $\mathbb{E}[\boldsymbol{\xi}_{a,i}|\mathbf{D}_1, \dots, \mathbf{D}_i, \mathbf{w}_0, \dots, \mathbf{w}_{i-1}] = 0$, as it can be written as

$$\mathbb{E}[\mathbb{E}[\boldsymbol{\xi}_{a,i}|\boldsymbol{\xi}_{1,i}, \mathbf{D}_1, \dots, \mathbf{D}_i, \mathbf{w}_0, \dots, \mathbf{w}_{i-1}]|\mathbf{D}_1, \dots, \mathbf{D}_i, \mathbf{w}_0, \dots, \mathbf{w}_{i-1}]] = 0.$$

### 3.3 Guarantees for Low-Precision Oja's Algorithm

Before presenting our main result, we present a general result that can apply to other noisy variants of Oja's rule and is of independent interest. The proof is deferred to Appendix Section D. Consider Oja's algorithm on matrices $\mathbf{A}_i \in \mathbb{R}_{d \times d}$, such that $\mathbf{A}_i = \eta \mathbf{D}_i + \boldsymbol{\Xi}_i$ where $\mathbf{D}_i$ are IID random matrices with $\mathbb{E}[\mathbf{D}_i] = \boldsymbol{\Sigma}$.

Let $\mathcal{S}_i$ be the set of all random vectors $\boldsymbol{\xi}$ in the first $i$ iterations of the algorithm and $\mathcal{F}_{i-}$ denote the $\sigma$-algebra generated by the random $\mathbf{D}_1, \dots, \mathbf{D}_i$ and $\mathcal{S}_{i-1}$. Define the operator $\mathbb{E}_i[.] := \mathbb{E}[.|\mathcal{F}_{i-}]$. We assume the noise term $\boldsymbol{\Xi}_i$ is measurable with respect to the filtration $\mathcal{F}_{i-}$ and unbiased conditioned on $\mathcal{F}_{i-}$, i.e., $\mathbb{E}_i[\boldsymbol{\Xi}_i|\mathcal{F}_{i-}] = \mathbf{0}_{d \times d}$. Let $\mathcal{V}_0, \nu, \mathcal{M}, \kappa$, and $\kappa_1$ be non-negative parameters such that

$$\max\left(\|\mathbb{E}[(\mathbf{D}_i - \boldsymbol{\Sigma})(\mathbf{D}_i - \boldsymbol{\Sigma})^T]\|, \|\mathbb{E}[(\mathbf{D}_i - \boldsymbol{\Sigma})^T(\mathbf{D}_i - \boldsymbol{\Sigma})]\|\right) \leqslant \mathcal{V}_0, \tag{8}$$

$$\|\mathbf{D}_i\| \leqslant 1, \qquad \|\mathbf{D}_i - \boldsymbol{\Sigma}\| \leqslant \mathcal{M}, \qquad \|\boldsymbol{\Xi}_i\| \leqslant \kappa, \qquad \|\mathbb{E}[\boldsymbol{\Xi}_i^T \boldsymbol{\Xi}_i|\mathcal{F}_{i-}]\|_F \leqslant \kappa_1 \quad \text{a.s.} \tag{9}$$

**Theorem 1.** *Let $d, n, b \in \mathbb{N}$ and $\mathbf{u}_0 \sim \mathcal{N}(0, \mathbf{I}_d)$. Let $\eta := \frac{\alpha \log n}{b(\lambda_1 - \lambda_2)}$ be the learning rate where $\alpha$ is chosen to satisfy Lemma A.2, and suppose $\max(b\eta^2 \mathcal{M}^2 \log(d), b\kappa^2 \log d) = O(1)$. Then, with probability at least 0.9, the vector $\mathbf{u}_b$ from equation 7 satisfies $\|\mathbf{u}_b\| \in [1 - \kappa_1, 1 + \kappa_1]$ and*

$$\sin^2(\mathbf{u}_b, \mathbf{v}_1) \lesssim \frac{d}{n^{2\alpha}} + \frac{\alpha \mathcal{V}_0 \log n}{b(\lambda_1 - \lambda_2)^2} + \max\left(\frac{b}{\alpha \log n}, 1\right) \kappa_1 + \kappa^2.$$

**Remark 1** (Matching the Upper and Lower Bounds). *In the LQ scheme with gap $\delta$, each coordinate of the noise vector $\boldsymbol{\xi}$ is bounded by $\delta$ almost surely. In particular, this implies $\kappa = O(\delta\sqrt{d})$ and $\kappa_1 = O(\delta^2 d)$ (see Appendix Section D) and the resulting error due to quantization matches the lower bound in Lemma 1. In the NLQ scheme with parameters $\zeta$ and $\delta_0$, the $i$th coordinate of the noise vectors $\xi$ is bounded by $\zeta|\mathbf{u}_i| + \delta_0$, where $\mathbf{u}$ is the vector being quantized. Since the vectors in consideration are bounded in norm by 1, this implies $\kappa = O(\zeta + \delta_0\sqrt{d})$ and $\kappa_1 = O(\zeta^2 + \delta_0^2 d)$ (see Appendix Section D). The resulting error matches the lower bound in Lemma 2 as long as the output vector has norm in the range $[1/2, 2]$.*

**Remark 2.** *Theorem 1 relies on the observation that accumulating the quantization error only $b$ times in Algorithm 1 leads to a smaller $\sin^2$ error. Moreover, choosing an appropriate batch size reduces the variance parameter $\mathcal{V}_0$ by a factor of $n/b$ because of averaging.*

**Remark 3** (Hyperparameters and eigengap). *The choice of the learning rate $\eta = \frac{\alpha \log n}{n(\lambda_1 - \lambda_2)}$ is also present in other works on streaming PCA [HP14, SOR14, Sha16a, Sha16b, AZL17, HNWTW20, JNN19, BDF13] to derive the statistically optimal sample complexity (up to logarithmic factors). If a smaller learning rate $\eta$ is used (for example, by using an upper bound $U$ on the eigengap $\lambda_1 - \lambda_2$), then the first error term of Theorem 1 will be larger, leading to a slightly larger sin-squared error. A similar argument applies to the choice of the batch size.*

**Remark 4** (Known $n$ in the learning rate). *The length of the stream $n$ is an input in Theorem 1, and the learning rate is constant over time. To handle variable learning rates using only constant-rate updates, a standard doubling trick [ACBFS95] can be used. Specifically, the time horizon is divided into blocks that double in size: the kth block has size $2^{k-1}$ and Oja's algorithm run on that block uses a learning rate corresponding to that block's size. When the algorithm run on this block terminates, the older estimate of the top eigenvector run on the previous block is replaced by this new estimate. This scheme effectively simulates a decaying learning rate while keeping the analysis tractable.*

### 3.4 Choosing the Optimal Quantization Parameters

To ensure a fair comparison between the linear and logarithmic quantization schemes, we fix a budget $\beta$ for the total number of bits used by the low-precision model. Moreover, our algorithms require that numbers in, say, $(-2, 2)$ are representable by the quantization scheme. Therefore, we must ensure that the upper and lower limits of the scheme cover this range.

The largest number representable in the linear quantization scheme is $\delta(2^\beta - 1)$ and the smallest negative number representable is $-\delta \cdot 2^\beta$. We choose $\delta = 2^{2-\beta}$, which covers the range $(-2, 2)$.

To motivate the choice of $\zeta$ and $\delta_0$, we note that the floating point scheme is a *discretization* of the logarithmic quantization scheme. The parameter $\delta_0$ in the logarithmic scheme represents the smallest representable positive real, which in the FPQ scheme is equal to $4 \cdot 2^{-2^{\beta_e - 1}}$, where $\beta_e$ is the number of bits used to represent the exponent. The parameter $\zeta$ represents *multiplicative* growth between adjacent quanta and is analogous to $2^{-\beta_m}$ in the FPQ scheme, where $\beta_m$ is the number of bits to represent the mantissa, and $\beta = \beta_m + \beta_e$. Assuming $\zeta = 2^{-\beta_m}$ and $\delta_0 = 4 \cdot 2^{-2^{\beta_e - 1}}$, where $\beta_m$ and $\beta_e$ are positive integers, the largest representable number is

$$q_{2^{\beta-1}} = \left( (1+\zeta)^{2^{\beta-1}} - 1 \right) \cdot \frac{\delta_0}{\zeta} \geqslant 2^{\beta_m - 1}.$$

To represent numbers in $(-2, 2)$, it suffices to ensure $\beta_m \geqslant 3$. This allows some freedom to select $\beta_m$ and $\beta_e$ such that the factor $\kappa_1 = \zeta^2 + \delta_0^2 d$ is minimized. We choose

$$\beta_e = \lceil \log_2 \left( 2\beta + \log_2(8d \ln 2) \right) \rceil \quad \text{and} \quad \beta_m = \beta - \beta_e$$

which is valid as long as $\beta \geqslant \max(8, \log_2 d)$ and $\beta_m \geqslant 3$. We justify this choice in appendix D.3.

With this choice of $\beta_e$ and $\beta_m$, the parameters $\zeta$ and $\delta_0$ satisfy

$$\delta_0^2 \leqslant \frac{2}{4^\beta d \ln 2} \quad \text{and} \quad \zeta^2 \leqslant \frac{4 \left( 2\beta + \log_2(8d \ln 2) \right)^2}{4^\beta}. \tag{10}$$

With this setting, we present two immediate corollaries of Theorem 1 with a fixed budget $\beta$. The proofs are deferred to Appendix Section D.

**Theorem 2.** *[Oja's Algorithm with Batches]*

1. *Suppose $\mathcal{Q} = \mathcal{Q}_L$ and $\delta, b$ satisfy $\delta = 2^{2-\beta} = O\left( \frac{\lambda_1 - \lambda_2}{\alpha \sqrt{d} \log(n)} \right)$ and $b = \Theta\left( \frac{\alpha^2 \log^2(n)}{(\lambda_1 - \lambda_2)^2} \right)$. Then, with probability at least $0.9$, the output $\mathbf{w}_b$ of Algorithm 1 satisfies*

$$\sin^2(\mathbf{w}_b, \mathbf{v}_1) \lesssim \frac{d}{n^{2\alpha}} + \frac{\alpha \log(n)}{(\lambda_1 - \lambda_2)^2} \left( \frac{\mathcal{V}}{n} + \frac{d}{4^\beta} \right).$$

2. *Suppose $\mathcal{Q} = \mathcal{Q}_{NL}$ with $\zeta$ and $\delta_0$ as in equation (10), such that $\zeta + \delta_0 \sqrt{d} = O\left( \frac{\lambda_1 - \lambda_2}{\alpha \sqrt{d} \log(n)} \right)$, and batch size $b = \Theta\left( \frac{\alpha^2 \log^2(n)}{(\lambda_1 - \lambda_2)^2} \right)$. Then, with probability at least $0.9$, the output $\mathbf{w}_b$ of Algorithm 1 satisfies*

$$\sin^2(\mathbf{w}_b, \mathbf{v}_1) \lesssim \frac{d}{n^{2\alpha}} + \frac{\alpha \log(n)}{(\lambda_1 - \lambda_2)^2} \left( \frac{\mathcal{V}}{n} + \frac{\beta^2 + \log^2(d)}{4^\beta} \right).$$

**Theorem 3.** *[Oja's Algorithm]*

1. *Suppose* $\mathcal{Q} = \mathcal{Q}_L$, *and* $\delta, b$ *satisfy* $\delta = 2^{2-\beta} = O\left(\min\left(\frac{\lambda_1 - \lambda_2}{\alpha\sqrt{d}\log(n)}, \frac{1}{\sqrt{dn}}\right)\right)$ *and* $b = n$. *Then, with probability at least* $0.9$, *the output* $\mathbf{w}_n$ *of Algorithm 1 satisfies*

$$\sin^2(\mathbf{w}_n, \mathbf{v}_1) \lesssim \frac{d}{n^{2\alpha}} + \frac{\alpha\mathcal{V}\log(n)}{n(\lambda_1 - \lambda_2)^2} + \frac{dn}{4^\beta\alpha\log(n)}.$$

2. *Suppose* $\mathcal{Q} = \mathcal{Q}_{NL}$ *with* $\zeta$ *and* $\delta_0$ *as in equation* (10), *such that* $\zeta + \delta_0\sqrt{d} < O\left(\min\left(\frac{\lambda_1 - \lambda_2}{\alpha\sqrt{d}\log(n)}, \frac{1}{\sqrt{dn}}\right)\right)$, *and batch size* $b = n$. *Then, with probability at least* $0.9$, *the output* $\mathbf{w}_n$ *of Algorithm 1 satisfies*

$$\sin^2(\mathbf{w}_n, \mathbf{v}_1) \lesssim \frac{d}{n^{2\alpha}} + \frac{\alpha\mathcal{V}\log(n)}{n(\lambda_1 - \lambda_2)^2} + \frac{(\beta^2 + \log^2 d)n}{4^\beta\alpha\log(n)}.$$

Under linear quantization (LQ), the quantization error term scales as $d/4^\beta$, whereas under nonlinear/logarithmic quantization (NLQ) it is only $(\beta^2 + \log^2 d)/4^\beta$. Thus, NLQ achieves a *nearly dimension-independent error* resulting from quantization, making it especially advantageous in high-dimensional settings.

The errors of Oja's algorithm with batching due to quantization are $\tilde{O}(d4^{-\beta})$ and $\tilde{O}(4^{-\beta})$ in the two cases of linear and logarithmic quantization, which are an $n$ factor larger than the corresponding errors without batching. Theorem 2 and 3 show that batching significantly improves the performance under quantization. They further show that the NLQ scheme, when suitably optimized, gives nearly dimension-independent dependence on the quantization error. In comparison, the error resulting from quantization in LQ suffers the most from higher dimensions. In Figure 1 we see that unquantized algorithms (standard and batched) have similar and best performance. See Section 5 for detailed experimental evidence supporting the theory.

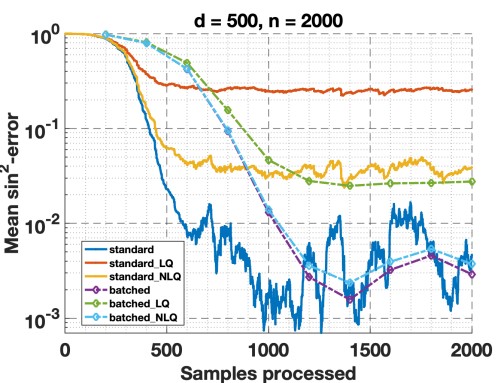

Figure 1: We study the effect of different quantization strategies on mean $\sin^2$-error over 10 runs as the number of samples grows on the $x$ axis. *Standard* uses $b = n$ batches whereas *Batched* uses $b = 10$ batches. Among the quantization algorithms, we see that in $\sin^2$ error, Standard LQ > Batched LQ and Standard NLQ > Batched NLQ.

**Remark 5.** *Theorems 2 and 3 are stated with a constant probability of success. In Section 3.5 we provide a quantized probability boosting algorithm (Algorithm 2) which boosts the probability of success from a constant to* $1 - \theta$ *for arbitrary* $\theta \in (0, 1]$.

### 3.5 Boosting the Probability of Success

Quantized Oja's algorithm produces an estimate whose error is within the target threshold with constant success probability. This section addresses this gap by presenting a standard probability boosting framework to let the failure probability $\theta$ be arbitrarily small.

Algorithm 2 begins by partitioning $m$ data $\{\mathbf{X}_i\}_{i\in[m]}$ into $r = \Theta(\log 1/\theta)$ disjoint batches of size $n$ each and runs the algorithm $\mathcal{A}$ on each batch. The output vectors $\{\mathbf{u}_i\}_{i\in[r]}$ are then aggregated using the boosting procedure SuccessBoost. This procedure looks for a *popular* vector $\mathbf{u}_i$ close to at least half of the other vectors and returns any such vector. A general argument for SuccessBoost for arbitrary distance metrics can be found in [KLL+23, KS24a].

---

**Algorithm 2** Probability Boosted Oja's Algorithm

---

**Require:** Data $\{\mathbf{X}_i\}_{i\in[m]}$, algorithm $\mathcal{A}$, quantization grid $\mathcal{Q}_L(\epsilon)$, failure probability $\theta$, error $\epsilon$

1: $r \leftarrow \lceil 20\log(1/\theta) \rceil, \quad n \leftarrow \lfloor m/r \rfloor$
2: **for** $i = 1$ to $r$ **do**
3: $\quad B_i \leftarrow \{(i-1)n, (i-1)n+1, \ldots, (i-1)n+n\}$
4: $\quad \mathbf{u}_i \leftarrow \mathcal{A}\left(\{\mathbf{X}_j\}_{j\in B_i}\right)$
5: **procedure** $\tilde{\rho}(\mathbf{x}, \mathbf{y})$
6: $\quad$ **return** $\mathsf{Q}\left(\sin^2(\mathbf{x}, \mathbf{y}), \mathcal{Q}_L(\epsilon)\right)$
7: **procedure** SuccessBoost($\{\mathbf{u}_i\}_{i\in[r]}, \rho, \epsilon$)
8: $\quad$ **for** $i = 1$ to $r$ **do**
9: $\quad\quad c_i \leftarrow |\{j \in [r] : \rho(\mathbf{u}_i, \mathbf{u}_j) \leqslant 5\epsilon\}|$
10: $\quad\quad$ **if** $c_i \geqslant 0.5r$ **then**
11: $\quad\quad\quad$ **return** $\mathbf{u}_i$
$\quad\quad$ **return** $\perp$
12: $\bar{\mathbf{u}} \leftarrow$ SuccessBoost($\{\mathbf{u}_i\}_{i\in[r]}, \tilde{\rho}, \epsilon$)
13: **return** $\bar{\mathbf{u}}$

---

We use a quantized version $\tilde{\rho}$ as a proxy for the $\sin^2$ error in the SuccessBoost procedure. $\tilde{\rho}$ uses the linear quantization grid

$$\mathcal{Q}_L^{(\beta)}(\epsilon) = \{-2^{\beta-1}\epsilon, -(2^{\beta-1}-1)\epsilon, \ldots, -\epsilon, 0, \epsilon, \ldots, (2^{\beta-1}-1)\epsilon\}, \tag{11}$$

where the gap $\epsilon$ is set to the upper bound on the error guaranteed by Theorem 2 or Theorem 3 depending on the algorithm $\mathcal{A}$ in use.

Standard arguments for SuccessBoost apply when the error $\tilde{\rho}$ is either computed exactly. The difference in our setting is that we the error function $\tilde{\rho}$ is only approximately a metric and does not behave as intended if the computed value is outside the quantization range. To highlight the second point, consider the *unbounded* quantization grid

$$\mathcal{Q}_L^*(\epsilon) = \{k\epsilon : k \in \mathbb{Z}\}.$$

With this grid, $\left|\tilde{\rho}(\mathbf{x}, \mathbf{y}) - \sin^2(\mathbf{x}, \mathbf{y})\right|$ is bounded by $O(\epsilon)$ almost surely. We extend the argument to show that Lemma 3 holds even with the bounded grid $\mathcal{Q}_L(\epsilon) := \mathcal{Q}_L(\epsilon, \beta)$, which truncates values outside the range $\left[-2^{\beta-1}\epsilon, (2^{\beta-1}-1)\epsilon\right]$ to its endpoints. This requires a modest assumption that the number of bits $\beta \geqslant 4$, which is already assumed when optimizing the parameters in Section 3.4.

**Lemma 3.** *Let $d > 1, \beta \geqslant 4, \epsilon \in (0, 0.75), \theta \in (0, 1)$, and $r = \lceil 20\log(1/\theta) \rceil$. Let $\mathbf{v} \in \mathbb{R}^d$ be a unit vector and $\mathbf{u}_1, \mathbf{u}_2, \ldots, \mathbf{u}_r$ be independent random vectors such that $\Pr\left(\sin^2(\mathbf{u}_i, \mathbf{v}) \leqslant \epsilon\right) \geqslant 0.9$. Let $\tilde{\rho}$ be the function defined in Algorithm 2 with the quantization grid $\mathcal{Q}_L(\epsilon, \beta)$. Then, the vector $\bar{\mathbf{u}} := SuccessBoost\left(\{\mathbf{u}_i\}_{i\in[r]}, \tilde{\rho}, \epsilon\right)$ satisfies*

$$\Pr\left(\sin^2(\bar{\mathbf{u}}, \mathbf{v}) \leqslant 14\epsilon\right) \geqslant 1 - \theta.$$

The proof of Lemma 3 is in Appendix E.

Algorithm 2 has a constant overhead in the error compared to algorithm $\mathcal{A}$. The probability of success is amplified from 0.9 to $1 - \theta$. The number of samples needed to achieve the same error (up to constant factors) as $\mathcal{A}$ blows up only by a multiplicative factor $\Theta(\log 1/\theta)$. If algorithm $\mathcal{A}$ runs in $O(nd)$ time and $O(d)$ space, which is the case for Oja's algorithm and its batch variants, then Algorithm 2 takes $O(nd\log(1/\theta) + d\log^2(1/\theta))$ time and $O(d\log(1/\theta))$ space.

## 4 Proof Techniques

Our proof of Theorem 1 has three main parts. Let $\mathbf{Z}_b = \prod_{i=b}^{1}(\mathbf{I} + \mathbf{A}_i)$ where $\mathbf{A}_i := \eta\mathbf{D}_i + \Xi_i$ as described in equation (7). First, note that the sin-squared error can be written as $1 - \left(\mathbf{u}_b^\top\mathbf{v}_1\right)^2 = \|\mathbf{V}_\perp\mathbf{V}_\perp^\top\mathbf{Z}_b\mathbf{u}_0\|^2/\|\mathbf{Z}_b\mathbf{u}_0\|^2$. Using the one-step power method result shown in Lemma 6 from [JJK+16], for a fixed $\theta \in (0, 1)$, with probability atleast $1 - \theta$,

$$1 - \left(\mathbf{u}_b^\top\mathbf{v}_1\right)^2 \leqslant \frac{3\log(1/\theta)}{\theta^2}\frac{\text{Tr}\left(\mathbf{V}_\perp^\top\mathbf{Z}_b\mathbf{Z}_b^\top\mathbf{V}_\perp\right)}{\mathbf{v}_1^\top\mathbf{Z}_b\mathbf{Z}_b^\top\mathbf{v}_1}. \tag{12}$$

This makes our strategy clear for the subsequent proof. We bound the numerator by bounding $\mathbb{E}[\mathrm{Tr}(\mathbf{V}_\perp{}^\top \mathbf{Z}_b \mathbf{Z}_b^\top \mathbf{V}_\perp)]$ and applying Markov's inequality. For the denominator, we lower bound $\left\| \mathbf{Z}_b^\top \mathbf{v}_1 \right\|$ by decomposing it as

$$\|\mathbf{Z}_b^\top \mathbf{v}_1\| \geqslant \| (\mathbf{I} + \eta\boldsymbol{\Sigma})^b \mathbf{v}_1 \| - \|(\mathbf{Z}_b - (\mathbf{I} + \eta\boldsymbol{\Sigma})^b)^\top \mathbf{v}_1 \| \geqslant (1 + \eta\lambda_1)^b - \|\mathbf{Z}_b - (\mathbf{I} + \eta\boldsymbol{\Sigma})^b \| \quad (13)$$

and upper-bounding $\|\mathbf{Z}_b - (\mathbf{I} + \eta\boldsymbol{\Sigma})^b \|$. For both the numerator and the denominator, we use the following intermediate bound, which controls the $(p, q)$-norm for a random matrix $\mathbf{X}$ defined as $\|\mathbf{X}\|_{p,q} = \mathbb{E}[\|\mathbf{X}\|_p^q]^{1/q}$, where $\|\mathbf{X}\|_p$ represents the Schatten-$p$ norm.

**Proposition 1.** *Let the noise term $\boldsymbol{\Xi}$, defined in (9), be bounded as $\|\boldsymbol{\Xi}\| \leqslant \kappa$ almost surely. Under Assumption 1, for $\eta \in (0, 1)$, we have*

$$\|\|\mathbf{Z}_b\|\|_{p,q}^2 \leqslant \phi^b \exp(C_p b\gamma) \|\mathbf{Z}_0\|_p^2$$
$$\|\|\mathbf{Z}_b - (\mathbf{I} + \eta\boldsymbol{\Sigma})^b\|\|_{p,q}^2 \leqslant \phi^b (\exp(C_p b\gamma) - 1) \|\mathbf{Z}_0\|_p^2,$$

*where $\mathbf{Z}_0 = \mathbf{I}$, $\phi := (1 + \eta\lambda_1)^2$, $\gamma := 2(\eta^2\mathcal{M}^2 + \kappa^2)$, and $C_p := p - 1$.*

The proof of Proposition 1 adapts the arguments for matrix product concentration from [HNWTW20]. which also include results for a general sequence of matrices adapted to a suitable filtration.

From Proposition 1 with $q = 2$, $p = 2 + 2\log d$, we get

$$\mathbb{E}\left[\|\mathbf{Z}_b - (\mathbf{I} + \eta\boldsymbol{\Sigma})^b\|\right] \leqslant \|\|\mathbf{Z}_b - (\mathbf{I} + \eta\boldsymbol{\Sigma})^b\|\|_{p,2} \leqslant \sqrt{e^2 b\gamma\,(1 + 2\log(d))}\,(1 + \eta\lambda_1)^b.$$

This allows us to control the lower bound via Markov's inequality, by substituting in equation (13).

To control the numerator, we show the following result (Lemma 4),
**Lemma 4.** *Let Assumption 1 hold and let $\gamma := 2(\eta^2\mathcal{M}^2 + \kappa^2)$. If $b\gamma\,(1 + 2\log(d)) \leqslant 1$, then*

$$\mathbb{E}[\mathrm{Tr}(\mathbf{V}_\perp{}^\top \mathbf{Z}_b \mathbf{Z}_b^\top \mathbf{V}_\perp)] \leqslant \exp\left(2\eta b\lambda_1 + \eta^2 b\left(\mathcal{V}_0 + \lambda_1^2\right)\right) \left(\frac{d}{\exp\left(2\eta b\left(\lambda_1 - \lambda_2\right)\right)} + \frac{5\eta^2\mathcal{V}_0 + 5\kappa_1}{\eta\left(\lambda_1 - \lambda_2\right)}\right).$$

The proof of Lemma 4 follows Lemma 10 of [JJK+16] to show, for $\beta_t := \mathbb{E}[\mathrm{Tr}(\mathbf{V}_\perp{}^\top \mathbf{Z}_t \mathbf{Z}_t^\top \mathbf{V}_\perp)]$,

$$\beta_t \leqslant \left(1 + 2\eta\lambda_2 + \eta^2\left(\mathcal{V}_0 + \lambda_1^2\right)\right)\beta_{t-1} + \left(\eta^2\mathcal{V}_0 + \kappa_1\right)\mathbb{E}[\|\mathbf{Z}_{t-1}\|^2].$$

At this step, we deviate from their proof and appeal to Proposition 1 for bounding $\mathbb{E}[\|\mathbf{Z}_{t-1}\|^2]$. Setting $\phi := (1 + \eta\lambda_1)^2$, $\gamma := 2(\eta^2\mathcal{M}^2 + \kappa^2)$ and $p := \max(2, \sqrt{2\log d/(b\gamma)})$, we get

$$\mathbb{E}[\|\mathbf{Z}_b\|]^2 \leqslant \|\|\mathbf{Z}_b\|\|_{p,2}^2 \leqslant \phi^b \exp(C_p b\gamma) \|\mathbf{Z}_0\|_p^2 \leqslant (1 + \eta\lambda_1)^{2b} \exp\left(2pb\gamma\right).$$

Unrolling the recursion and using this bound proves Lemma 4. The proof of Theorem 1 then follows from the one-step power method guarantee in equation 12. Detailed proofs are in Appendix C.

## 5   Experiments

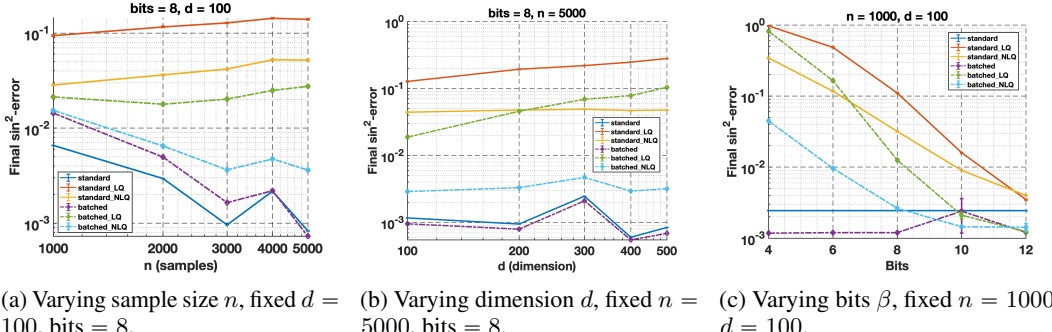

(a) Varying sample size $n$, fixed $d = 100$, bits $= 8$.

(b) Varying dimension $d$, fixed $n = 5000$, bits $= 8$.

(c) Varying bits $\beta$, fixed $n = 1000$, $d = 100$.

Figure 2: Variation of $\sin^2$-error with (a) sample size, (b) dimension, and (c) quantization bits.

We generate $n$ samples from a $d$ dimensional distribution selected by choosing a random orthonormal matrix $\boldsymbol{Q}$, setting $\boldsymbol{\Sigma} := \boldsymbol{Q}\boldsymbol{\Lambda}\boldsymbol{Q}^\top$ for $\boldsymbol{\Lambda}_{ii} := i^{-2}$ and sampling datapoints i.i.d from $\mathcal{N}(0, \boldsymbol{\Sigma})$. We compare six variants of Oja's algorithm for estimating $v_1$, the leading eigenvector of $\Sigma$. The baseline is the standard full precision update in Eq 1 (*standard*). *standard_LQ* and *standard_NLQ* use Algorithm 1 with $b = n$ and $Q(., \mathcal{Q}_L)$ and $Q(., \mathcal{Q}_{NL})$ respectively. The *batched* variant follows Eq 2 with $b = 100$ (for Figures 2a and 2b) and $b = 25$ (for Figure 2c) equal-sized batches. Finally, we combine the batched schedule by running Algorithm 1 with $\mathsf{Q}(., \mathcal{Q}_L)$ (*batched_LQ*) and with $\mathsf{Q}(., \mathcal{Q}_{NL})$ (*batched_NLQ*). All experiments were done on a personal computer with a single CPU.

The low-precision methods rely on Eq 10 to choose quantization parameters for a target number of bits $\beta = 8$. Given the dimension $d$, these routines compute a uniform quantization step $\delta_{\mathrm{uni}}$, an exponential step $\delta_{\mathrm{exp}}$, and a multiplicative-growth factor $\alpha_{\mathrm{exp}}$ to cover a fixed dynamic range. Each configuration is run for $R = 100$ independent trials. In Experiment 1 we fix $d = 100$ and vary $n \in \{1000, 2000, 3000, 4000, 5000\}$; in Experiment 2 we fix $n = 5000$ and vary $d \in \{100, 200, 300, 400, 500\}$. Every trial begins from a random Gaussian vector normalized to unit length. We set the learning rate to $\eta = \frac{2 \ln(n)}{n \, (\lambda_1 - \lambda_2)}$ for the standard method and to $\eta = \frac{2 \ln(n)}{b \, (\lambda_1 - \lambda_2)}$ for the batched methods. Upon completion we record the final excess error $\sin^2(\hat{\mathbf{w}}, \mathbf{v}_1) = 1 - (\hat{\mathbf{w}}^\top \mathbf{v}_1)^2$ and report the mean. The first two use the log-log scale and the third uses the log scale for the $y$-axis.

As shown in Figure 2a, all methods improve as the number of samples $n$ grows except *standard_LQ* and *standard_NLQ*. The errors of these two methods, as expected from Theorem 3, grow linearly with $n$. In contrast, the *batched_LQ* and *batched_NLQ*'s quantization errors do not depend linearly on $n$ and improve over the standard counterparts. Figure 2b shows how the error varies with the data dimension $d$. Since $\mathcal{V}$ grows mildly with $d$, for our data distribution, all methods other than *standard_LQ* and *batched_LQ* do not grow with $d$. These two methods grow linearly with $d$, confirming our theoretical findings in the first results under Theorems 2 and 3. Finally, Figure 2c compares the errors with the bit budget $\beta$. As $\beta$ increases from 4 to 12, linear and logarithmic quantization schemes steadily reduce their error and converge toward the full-precision result by $\beta = 12$. The batched quantizers require only 6–8 bits to achieve comparable performance to the full-precision batched error, whereas the *standard_LQ* and *standard_NLQ* need at least 10 bits to reach the same performance. The variability of the full precision methods arises from the randomness of initializations. Appendix F provides experiments on additional real-world and synthetic data.

## 6 Conclusion

We study the effect of linear (LQ) and logarithmic (NLQ) stochastic quantization on Oja's algorithm for streaming PCA. We obtain new lower bounds under both quantization settings and show that the batch variant of our quantized streaming algorithm achieves the lower bound up to logarithmic factors. The lower bound on the quantization error resulting from our logarithmic quantization is dimension-free. In contrast, the quantization error under the LQ scheme depends linearly in $d$, which is problematic in high dimensions. We also show a surprising phenomenon under quantization: the quantization error of standard Oja's algorithm scales with $n$ under both NLQ and LQ schemes, while batch updates with a small batch size does not incur this dependence. These theoretical observations are validated via experiments. A limitation of our analysis is that we estimate the first principal component only. Deflation-based approaches (see e.g. [JKL$^+$24, Mac08, SJS09]) provide an interesting future direction for extending this work for retrieving the top $k$ principal components.

## Acknowledgments and Disclosure of Funding

We gratefully acknowledge NSF grants 2217069, 2019844, and DMS 2109155.

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

## A    Utlity Results

**Lemma A.1.** *Let $l \leqslant x \leqslant u$ be reals, and define*

$$Q(x, \mathcal{Q}) = \begin{cases} \ell & \text{with probability } 1 - p(x) \\ u & \text{with probability } p(x) \end{cases},$$

*where $p(x) := (x - \ell)/(u - \ell)$. Then,*

(i) $\mathbb{E}\left[Q(x, \mathcal{Q})|x\right] = x$.

(ii) $|Q(x, \mathcal{Q}) - x| \leqslant u - l$.

(iii) $\mathsf{Var}\left[Q(x, \mathcal{Q})|x\right] \leqslant \frac{(u-l)^2}{4}$.

*Proof.* Throughout the proof, we condition on the fixed $x$ and treat all randomness as coming from the independent choices made by the quantizer.

(i) *Unbiasedness.* We have

$$\mathbb{E}[Q(x, \delta) \mid x] = p_i(x)u + (1 - p_i(x))\ell = x.$$

(ii) *Boundedness.* By definition, after rounding, we always round any $x \in [u, l]$ to either $u$ or $l$. Therefore, $|Q(x, \mathcal{Q}) - x| \leqslant u - l$.

(iii)*Variance bound.* Using the variance of a Bernoulli random variable, we have,

$$\mathsf{Var}[Q(x, \mathcal{Q}) \mid x] = p_i(x)(1 - p_i(x))(u - l)^2 \leqslant \frac{1}{4}(u - l)^2$$

since $t(1 - t) \leqslant 1/4$ for all reals $t$. $\qquad\square$

**Lemma A.2** (Choice of learning rate). *Let $\eta := \frac{\alpha \log(n)}{b(\lambda_1 - \lambda_2)}$. Then, under Assumption 1, for $\theta \in (0, 1)$, $\eta$ satisfies*

$$b(\eta^2 \mathcal{M}^2 + \kappa^2) \leqslant \frac{0.008}{\log(d/\theta)}, \text{ and } \eta \in (0, 1)$$

*for $\alpha > 1$, $b \geqslant 250\alpha^2 \log^2(n) \log\left(\frac{d}{\theta}\right) / (\lambda_1 - \lambda_2)^2$, and $\kappa^2 b \leqslant 0.004/\log\left(\frac{d}{\theta}\right)$.*

*Proof.* For Lemma A.8, we require,

$$4b(\eta^2 \mathcal{M}^2 + \kappa^2)(1 + 2\log(d)) \leqslant 1 \tag{A.14}$$

For Theorem A.4, we require,

$$4e^2 b(\eta^2 \mathcal{M}^2 + \kappa^2) \log\left(\frac{d}{\theta}\right) \leqslant \frac{1}{4} \tag{A.15}$$

where $\theta \in (0, 1)$ represents the failure probability. It is not hard to see that (A.15) implies (A.14). Therefore it suffices to ensure

$$b(\eta^2 \mathcal{M}^2 + \kappa^2) \log \left( \frac{d}{\theta} \right) \leqslant 0.008$$

Setting each term smaller than $0.004$, it suffices to have

$$b \geqslant \frac{250\alpha^2 \log^2(n) \log \left( \frac{d}{\theta} \right)}{(\lambda_1 - \lambda_2)^2}, \quad \kappa^2 b \leqslant \frac{0.004}{\log \left( \frac{d}{\theta} \right)}$$

which completes the proof for the first condition.

The second condition on $\eta$ follows by setting $\eta \leqslant 1$ and solving for $b$. This yields

$$b \geqslant \max \left\{ 250\alpha^2 \log^2(n) \log \left( \frac{d}{\theta} \right) / (\lambda_1 - \lambda_2)^2, \alpha \log(n) / (\lambda_1 - \lambda_2) \right\}$$

Since $\alpha > 1$, the first term is larger than the second one, which completes the proof. $\qquad\square$

**Lemma A.3.** *Let $\mathbf{w}$ and $\boldsymbol{\xi}$ be vectors in $\mathbb{R}^d$ such that $\|\mathbf{w}\| = 1$ and $\mathbf{w} + \boldsymbol{\xi} \neq 0$. Then,*

$$\sin^2(\mathbf{w}, \mathbf{w} + \boldsymbol{\xi}) \leqslant \left( \frac{\|\boldsymbol{\xi}\|}{\|\mathbf{w} + \boldsymbol{\xi}\|} \right)^2.$$

*Proof.*

$$\sin^2(\mathbf{w}, \mathbf{w} + \boldsymbol{\xi}) = 1 - \left( \frac{\mathbf{w}^\top(\mathbf{w} + \boldsymbol{\xi})}{\|\mathbf{w} + \boldsymbol{\xi}\|} \right)^2 = \frac{(\mathbf{w} + \boldsymbol{\xi})^\top(\mathbf{w} + \boldsymbol{\xi}) - (1 + \mathbf{w}^\top\boldsymbol{\xi})^2}{\|\mathbf{w} + \boldsymbol{\xi}\|^2}$$

$$= \frac{\boldsymbol{\xi}^\top\boldsymbol{\xi} - (\mathbf{w}^\top\boldsymbol{\xi})^2}{\|\mathbf{w} + \boldsymbol{\xi}\|^2} \leqslant \left( \frac{\|\boldsymbol{\xi}\|}{\|\mathbf{w} + \boldsymbol{\xi}\|} \right)^2.$$

$\square$

**Lemma A.4.** *Let $\mathbf{x}$ and $\mathbf{y}$ be unit vectors in $\mathbb{R}^d$. Then,*

$$\frac{1}{2} \min(\|\mathbf{x} - \mathbf{y}\|^2, \|\mathbf{x} + \mathbf{y}\|^2) \leqslant \sin^2(\mathbf{x}, \mathbf{y}) \leqslant \min(\|\mathbf{x} - \mathbf{y}\|^2, \|\mathbf{x} + \mathbf{y}\|^2).$$

*Proof.* We express $\sin^2(\mathbf{x}, \mathbf{y})$ in terms of $\|\mathbf{x} - \mathbf{y}\|$ and $\|\mathbf{x} + \mathbf{y}\|$. Since $\|\mathbf{x} - \mathbf{y}\|^2 = \|\mathbf{x}\|^2 + \|\mathbf{y}\|^2 - 2\mathbf{x}^\top\mathbf{y} = 2 - 2\cos(\mathbf{x}, \mathbf{y})$ and $\|\mathbf{x} + \mathbf{y}\|^2 = 2 + 2\cos(\mathbf{x}, \mathbf{y})$,

$$\|\mathbf{x} - \mathbf{y}\|^2 + \|\mathbf{x} + \mathbf{y}\|^2 = 4 \text{ and } \sin^2(\mathbf{x}, \mathbf{y}) = 1 - \cos^2(\mathbf{x}, \mathbf{y}) = \frac{1}{4} \|\mathbf{x} - \mathbf{y}\|^2 \|\mathbf{x} + \mathbf{y}\|^2.$$

The upper bound on $\sin^2(\mathbf{x}, \mathbf{y})$ follows immediately from the above equations. For the lower bound, note that at least one of $\|\mathbf{x} - \mathbf{y}\|^2$ and $\|\mathbf{x} + \mathbf{y}\|^2$ is at least $2$ because their sum is equal to $4$. If $\|\mathbf{x} + \mathbf{y}\|^2 \geqslant 2$, then $\sin^2(\mathbf{x}, \mathbf{y}) \geqslant \|\mathbf{x} - \mathbf{y}\|^2 / 2$. Otherwise, $\sin^2(\mathbf{x}, \mathbf{y}) \geqslant \|\mathbf{x} + \mathbf{y}\|^2 / 2$. $\qquad\square$

**Lemma A.5.** *Let $\mathbf{x}, \mathbf{y}$, and $\mathbf{z}$ be non-zero vectors in $\mathbb{R}^d$. Then,*

$$\sin^2(\mathbf{x}, \mathbf{z}) \leqslant 2\sin^2(\mathbf{x}, \mathbf{y}) + 2\sin^2(\mathbf{y}, \mathbf{z}).$$

*Proof.* For unit vectors $\mathbf{u}$ and $\mathbf{v}$ in $\mathbb{R}^d$,

$$\left\| \mathbf{u}\mathbf{u}^\top - \mathbf{v}\mathbf{v}^\top \right\|_F^2 = \text{Tr}\left( (\mathbf{u}\mathbf{u}^\top - \mathbf{v}\mathbf{v}^\top)^2 \right)$$
$$= \text{Tr}\left( \mathbf{u}\mathbf{u}^\top - (\mathbf{u}^\top\mathbf{v})\mathbf{u}\mathbf{v}^\top - (\mathbf{v}^\top\mathbf{u})\mathbf{v}\mathbf{u}^\top + \mathbf{v}\mathbf{v}^\top \right)$$
$$= 2 - 2(\mathbf{u}^\top\mathbf{v})^2 = 2\sin^2(\mathbf{u}, \mathbf{v}).$$

By parallelogram law,

$$\frac{1}{2} \left\| \mathbf{x}\mathbf{x}^\top - \mathbf{z}\mathbf{z}^\top \right\|_F^2 \leqslant \left\| \mathbf{x}\mathbf{x}^\top - \mathbf{y}\mathbf{y}^\top \right\|_F^2 + \left\| \mathbf{y}\mathbf{y}^\top - \mathbf{z}\mathbf{z}^\top \right\|_F^2$$
$$\implies \sin^2(\mathbf{x}, \mathbf{z}) \leqslant 2\sin^2(\mathbf{x}, \mathbf{y}) + 2\sin^2(\mathbf{y}, \mathbf{z}).$$

$\square$

# B  Lower Bounds

## Proof of Lemma 1

*Proof.* Let $\mathbf{v}_1 \in \mathbb{R}^d$ be the unit vector with $\mathbf{v}_1(i) = \delta/3$ for $i \in [d-1]$ and $\mathbf{v}_1(d) = \sqrt{1 - \frac{(d-1)\delta^2}{9}}$.

Consider any a vector $\mathbf{w} \in \mathcal{V}_L$, and let $\tilde{\mathbf{w}} = \mathbf{w}/\|\mathbf{w}\|$. Since $\mathbf{w} \in \mathcal{V}_L$, $\mathbf{w}(i) = 0$ or $|\mathbf{w}(i)| \geqslant \delta/2$. In particular, $|\mathbf{v}_1(i) - \mathbf{w}(i)| \geqslant \delta/6$ and $|\mathbf{v}_1(i) + \mathbf{w}(i)| \geqslant \delta/6$ for all $i \in [d-1]$. It follows that

$$\|\mathbf{v}_1 - \mathbf{w}\|^2 \geqslant \sum_{i=1}^{d-1} (\mathbf{v}_1(i) - \mathbf{w}(i))^2 \geqslant (d-1) \left(\frac{\delta}{6}\right)^2 = \frac{\delta^2(d-1)}{36}$$

and $\|\mathbf{v}_1 + \mathbf{w}\|^2 \geqslant \frac{\delta^2(d-1)}{36}$ similarly. The Lemma follows from A.4.

$\square$

## Proof of Lemma 2

*Proof.* It suffices to construct two unit vectors $\mathbf{v}_1$ and $\mathbf{v}_2$ such that $\inf_{\mathbf{w} \in \mathcal{V}_{NL}} \sin^2(\mathbf{w}, \mathbf{v}_1) = \Omega(\zeta^2)$ and $\inf_{\mathbf{w} \in \mathcal{V}_{NL}} \sin^2(\mathbf{w}, \mathbf{v}_2) = \Omega(\delta_0^2 d)$.

Let $\mathbf{v}_1$ be the vector in $\mathbb{R}^d$ with coordinates

$$\mathbf{v}_1(1) = \frac{1}{\sqrt{1 + (1 + \zeta/2)^2}}, \quad \mathbf{v}_1(2) = \frac{1 + \zeta/2}{\sqrt{1 + (1 + \zeta/2)^2}}, \quad \mathbf{v}_1(i) = 0 \ \forall \ i \geqslant 3.$$

For the sake of contradiction, suppose there exists $\mathbf{w}_1 \in \mathcal{V}_{NL}$ such that $\sin^2(\mathbf{w}_1, \mathbf{v}_1) \leqslant \zeta^2/100$. Let $\tilde{\mathbf{w}}_1 := \mathbf{w}_1/\|\mathbf{w}_1\|$. By Lemma A.4,

$$\min(\|\mathbf{v}_1 - \tilde{\mathbf{w}}_1\|_2^2, \|\mathbf{v}_1 + \tilde{\mathbf{w}}_1\|_2^2) \leqslant 2\sin^2(\mathbf{v}_1, \mathbf{w}_1) \leqslant \frac{\zeta^2}{50}.$$

Flipping the sign of $\mathbf{w}_1$ if necessary, we may assume $\|\mathbf{v}_1 - \tilde{\mathbf{w}}_1\|_2^2 \leqslant \zeta^2/50$. So,

$$|\mathbf{v}_1(i) - \tilde{\mathbf{w}}_1(i)| \leqslant \zeta/7 \ \forall \ i \in [d]. \tag{A.16}$$

The bound $\zeta \leqslant 0.1$ ensures $\mathbf{v}_1(1) \geqslant 20/29$ and $\mathbf{v}_1(2) - \mathbf{v}_1(1) \geqslant \zeta/3$, which also implies $\tilde{\mathbf{w}}_1(2) - \tilde{\mathbf{w}}_1(1) \geqslant \zeta/3 - 2\zeta/7 = \zeta/21 > 0$. It follows that

$$\begin{aligned}
\frac{\mathbf{w}_1(2) + \delta_0/\zeta}{\mathbf{w}_1(1) + \delta_0/\zeta} &= \frac{\tilde{\mathbf{w}}_1(2) + \delta_0/\zeta \cdot 1/\|\mathbf{w}_1\|}{\tilde{\mathbf{w}}_1(1) + \delta_0/\zeta \cdot 1/\|\mathbf{w}_1\|} \leqslant \frac{\mathbf{v}_1(2) + \zeta/7 + \delta_0/2\zeta}{\mathbf{v}_1(1) - \zeta/7 + \delta_0/2\zeta} \\
&= 1 + \frac{\zeta}{2} + \frac{\delta_0/2\zeta + \zeta/7 - (1 + \zeta/2)(\delta_0/2\zeta - \zeta/7)}{\mathbf{v}_1(1) + \delta_0/2\zeta - \zeta/7} \\
&= 1 + \frac{\zeta}{2} + \frac{2\zeta/7 + \zeta^2/14 - \delta_0/4}{\mathbf{v}_1(1) - \zeta/7 + \delta_0/2\zeta} \\
&\leqslant 1 + \frac{\zeta}{2} + \frac{2\zeta/7}{2/3} < 1 + \zeta,
\end{aligned}$$

and

$$\begin{aligned}
\frac{\mathbf{w}_1(2) + \delta_0/\zeta}{\mathbf{w}_1(1) + \delta_0/\zeta} &= \frac{\tilde{\mathbf{w}}_1(2) + \delta_0/\zeta \cdot 1/\|\mathbf{w}_1\|}{\tilde{\mathbf{w}}_1(1) + \delta_0/\zeta \cdot 1/\|\mathbf{w}_1\|} \geqslant \frac{\mathbf{v}_1(2) - \zeta/7 + 2\delta_0/\zeta}{\mathbf{v}_1(1) + \zeta/7 + 2\delta_0/\zeta} \\
&= 1 + \frac{\zeta}{2} + \frac{2\delta_0/\zeta - \zeta/7 - (1 + \zeta/2)(2\delta_0/\zeta + \zeta/7)}{\mathbf{v}_1(1) + \zeta/7 + 2\delta_0/\zeta} \\
&= 1 + \frac{\zeta}{2} - \frac{2\zeta/7 + \zeta^2/14 + \delta_0}{\mathbf{v}_1(1) + \zeta/7 + 2\delta_0/\zeta} \\
&> 1 + \frac{\zeta}{2} - \frac{\zeta\mathbf{v}_1(1)/2 + \zeta^2/14 + \delta_0}{\mathbf{v}_1(1) + \zeta/7 + 2\delta_0/\zeta} = 1.
\end{aligned}$$

Under the logarithmic quantization scheme, it can be inductively shown that

$$q_k + \delta_0/\zeta = (\delta_0/\zeta) \cdot (1 + \zeta)^k$$

for all non-negative integers $k$ such that $q_k \in \mathcal{Q}_{NL}$. In particular, $\frac{\mathbf{w}_1(2)+\delta_0/\zeta}{\mathbf{w}_1(1)+\delta_0/\zeta}$ must be an integral power of $1 + \zeta$, contradicting

$$1 < \frac{\mathbf{w}_1(2) + \delta_0/\zeta}{\mathbf{w}_1(1) + \delta_0/\zeta} < 1 + \zeta.$$

Therefore, $\inf_{\mathbf{w}_1 \in \mathcal{V}_{NL}} \sin^2(\mathbf{w}_1, \mathbf{v}_1) \geqslant \zeta^2/100$.

The other bound is similar to the linear case. let $\mathbf{v}_2$ be the vector with coordinates

$$\mathbf{v}_2(d) = \sqrt{1 - (d-1)\delta_0^2/9}, \quad \mathbf{v}_1(i) = \frac{\delta_0}{3} \ \forall \ i \leqslant d - 1.$$

Any $\mathbf{w}_2 \in \mathcal{V}_{NL}$ satisfies $\mathbf{w}_2(i) = 0$ or $|\mathbf{w}_2(i)| \geqslant \delta_0$ for all $i \in [d]$. Since $\|\mathbf{w}_2\| \in [1/2, 2]$, the normalized vector $\tilde{\mathbf{w}}_2 = \mathbf{w}_2/\|\mathbf{w}_2\|$ satisfies $|\tilde{\mathbf{w}}_2(i)| = 0$ or $|\tilde{\mathbf{w}}_2(i)| \geqslant \delta_0/2$ for all $i \in [d]$.

In particular $|\mathbf{v}_2(i) - \tilde{\mathbf{w}}_2(i)| \geqslant \delta_0/6$ and $|\mathbf{v}_2(i) + \tilde{\mathbf{w}}_2(i)| \geqslant \delta_0/6$ for all $i \in [d]$. By Lemma A.4,

$$\sin^2(\mathbf{w}_2, \mathbf{v}_2) \geqslant \frac{1}{2} \min\left( \|\mathbf{w}_2 - \mathbf{v}_2\|^2, \|\mathbf{w}_2 + \mathbf{v}_2\|^2 \right) \geqslant \frac{\delta_0^2(d-1)}{72}.$$

$\square$

## C  Proof of Results in Section 4

For ease of exposition, all results in this section are stated with a generic number of data $n$. We apply these results with different choices of $n$ (e.g. number of batches $b$) for proving the main theorems (Theorem 1, 2, 3). Consider Oja's Algorithm applied to the matrices $\mathbf{A}_i \in \mathbb{R}_{d \times d}$, such that $\mathbf{A}_i = \eta \mathbf{D}_i + \boldsymbol{\Xi}_i$ where $\mathbf{D}_i$ are independent with $\mathbb{E}[\mathbf{D}_i] = \boldsymbol{\Sigma}$. Let $\mathcal{S}_i$ be the set of all random vectors $\boldsymbol{\xi}$ resulting from the quantizations in the first $i$ iterations of the algorithm, and let $\mathcal{F}_{i-}$ denote the $\sigma$-field generated by $\mathbf{D}_1, \ldots, \mathbf{D}_i$ and $\mathcal{S}_{i-1}$, and denote $\mathbb{E}_i[.] := \mathbb{E}[.|\mathcal{F}_{i-}]$. We assume the noise term $\boldsymbol{\Xi}_i$ is conditionally unbiased, i.e., $\mathbb{E}_i[\boldsymbol{\Xi}_i] = \mathbf{0}_{d \times d}$.

$$\mathcal{F}_{i-} := \sigma\left(\{\mathbf{D}_1, \ldots \mathbf{D}_i, \mathcal{S}_{i-1}\}\right), \qquad \mathcal{F}_i := \sigma\left(\{\mathbf{D}_1, \ldots \mathbf{D}_i, \mathcal{S}_i\}\right).$$

Recall the update rule

$$\mathbf{u}_i = (\mathbf{I} + \mathbf{A}_i)\mathbf{w}_{i-1}; \qquad \mathbf{w}_i = \frac{\mathbf{u}_i}{\|\mathbf{u}_i\|} = \frac{\prod_{t=i}^1 (\mathbf{I} + \mathbf{A}_t)\mathbf{u}_0}{\|\prod_{t=i}^1 (\mathbf{I} + \mathbf{A}_t)\mathbf{u}_0\|}. \tag{A.17}$$

We bound the numerator and denominator in (A.17) separately.

For the numerator, we will show that $\|\prod_{t=n}^1 (\mathbf{I} + \mathbf{A}_t) - (\mathbf{I} + \eta\boldsymbol{\Sigma})^n\|$ is small. Let $\mathbf{Y}_i = \mathbf{I} + \mathbf{A}_i$ for $i \in [n]$, and let $\{\mathbf{Z}_i\}_{0 \leqslant i \leqslant n}$ be defined as

$$\mathbf{Z}_i := \mathbf{Y}_i \mathbf{Z}_{i-1}, \qquad \mathbf{Z}_0 := \mathbf{I}. \tag{A.18}$$

Note that $\mathbf{Z}_{i-1}$ is measurable w.r.t $\mathcal{F}_{i-}$.

We are now ready to state our first result. Note that

$$\mathbf{Z}_n = \prod_{i=n}^1 (\mathbf{I} + \mathbf{A}_i).$$

where $\mathbf{A}_i = \eta \mathbf{D}_i + \boldsymbol{\Xi}_i$ and $\mathbf{D}_i$ are independent $d \times d$ random matrices with mean $\boldsymbol{\Sigma}$.

### C.1 Proof of Proposition 1

**Proposition A.1.** *[Proposition 1 in main paper]Let the noise term $\boldsymbol{\Xi}$, defined in* (9)*, be bounded as* $\|\boldsymbol{\Xi}\| \leqslant \kappa$ *almost surely. Under Assumption 1, for $\eta \in (0,1)$ and $b > 0$, we have*

$$\|\mathbf{Z}_b\|_{p,q}^2 \leqslant \phi^b \exp(C_p b \gamma) \|\mathbf{Z}_0\|_p^2$$

$$\|\mathbf{Z}_b - (\mathbf{I} + \eta \boldsymbol{\Sigma})^b\|_{p,q}^2 \leqslant \phi^b (\exp(C_p b \gamma) - 1) \|\mathbf{Z}_0\|_p^2,$$

*where $\mathbf{Z}_0 = \mathbf{I}$, $\phi := (1 + \eta \lambda_1)^2$, $\gamma := 2(\eta^2 \mathcal{M}^2 + \kappa^2)$, and $C_p := p - 1$.*

*Proof.* Recall the notation $\mathbf{Y}_i := \mathbf{I} + \mathbf{A}_i$ for all $i$. Then,

$$\mathbb{E}[\mathbf{Y}_i | \mathcal{F}_{i-1}] = \mathbf{I} + \eta \boldsymbol{\Sigma} + \mathbb{E}[\mathbb{E}_{i-}[\boldsymbol{\Xi}_i] | \mathcal{F}_{i-1}] = \mathbf{I} + \eta \boldsymbol{\Sigma}$$

Note that $m_i = 1 + \eta \lambda_1$ and

$$\|\mathbf{Y}_i - \mathbb{E}[\mathbf{Y}_i | \mathcal{F}_{i-1}]\| = \|\eta(\mathbf{D}_i - \boldsymbol{\Sigma}) + \boldsymbol{\Xi}_i\| \leqslant \eta \mathcal{M} + \kappa$$

The last line uses Eq 9. Thus $\sigma_i = \frac{\eta \mathcal{M} + \kappa}{1 + \eta \lambda_1}$. Note that $\nu \leqslant 2(\eta^2 \mathcal{M}^2 + \kappa^2)$. The same argument as in Theorem 7.4 in [HNWTW20] gives the bound. $\square$

**Lemma A.6.** *Under Assumption 1, and with $\eta$ set according to Lemma A.2 with $b = n$,*

$$\mathbb{P}\left(\|\mathbf{Z}_n - (\mathbf{I} + \eta \boldsymbol{\Sigma})^n\| \geqslant t (1 + \eta \lambda_1)^n\right) \leqslant \max(d, e) \exp\left(-\frac{t^2}{2e^2 n \gamma}\right) \quad \forall\ t \leqslant e.$$

*where $\gamma := 2(\eta^2 \mathcal{M}^2 + \kappa^2)$ and $e = \exp(1)$ is the Napier's constant.*

*Proof.* By Proposition A.1, for any positive real $p$,

$$\mathbb{P}\left(\|\mathbf{Z}_n - (\mathbf{I} + \eta \boldsymbol{\Sigma})^n\| \geqslant t (1 + \eta \lambda_1)^n\right) \leqslant \frac{\mathbb{E}\left[\|\mathbf{Z}_n - (\mathbf{I} + \eta \boldsymbol{\Sigma})^n\|^p\right]}{t^p (1 + \eta \lambda_1)^p} \leqslant \frac{\|\mathbf{Z}_n - (\mathbf{I} + \eta \boldsymbol{\Sigma})^n\|_{p,p}^p}{t^p (1 + \eta \lambda_1)^p}$$

$$\leqslant \frac{\phi^{\frac{p}{2}} (\exp(C_p n \gamma) - 1)^{p/2} d}{t^p (1 + \eta \lambda_1)^p} \leqslant d \left(t^{-2} (\exp(C_p n \gamma) - 1)\right)^{p/2},$$

where $\phi = (1 + \eta \lambda_1)^2$, $\gamma = 2(\eta^2 \mathcal{M}^2 + \kappa^2)$, and $C_p = p - 1$.

If $\frac{t^2}{e^2 n \gamma} < 2$, then $e \cdot \exp\left(-\frac{t^2}{2e^2 n \gamma}\right) \geqslant 1$ and the Lemma holds trivially. Otherwise, let $p := \frac{t^2}{e^2 n \gamma} \geqslant 2$. Since $t \leqslant e$, $C_p n \gamma \leqslant p n \gamma \leqslant \frac{t^2}{e^2} \leqslant 1$. Therefore, $\exp(C_p n \gamma) - 1 \leqslant e C_p n \gamma \leqslant \frac{t^2}{e}$, which implies

$$\mathbb{P}\left(\|\mathbf{Z}_n - (\mathbf{I} + \eta \boldsymbol{\Sigma})^n\| \geqslant t (1 + \eta \lambda_1)^n\right) \leqslant d \left(t^{-2} \cdot \frac{t^2}{e}\right)^{p/2} = d \exp\left(-\frac{t^2}{2e^2 n \gamma}\right).$$

$\square$

**Lemma A.7.** *Under Assumption 1 and with $\eta$ set according to Lemma A.2 with $b = n$,*

$$\mathbb{E}\left[\|\mathbf{Z}_n\|^2\right] \leqslant \exp\left(2\sqrt{2n\gamma \max\{2n\gamma, \log(d)\}}\right) (1 + \eta \lambda_1)^{2n},$$

*where $\gamma = 2(\eta^2 \mathcal{M}^2 + \kappa^2)$. Moreover, if $2n\gamma (1 + 2\log(d)) \leqslant 1$, then*

$$\mathbb{E}\left[\|\mathbf{Z}_n - \mathbb{E}[\mathbf{Z}_n]\|^2\right] \leqslant 2e^2 n\gamma (1 + 2\log(d)) (1 + \eta \lambda_1)^{2n}.$$

*Proof.* Using Proposition A.1 $\phi := (1 + \eta \lambda_1)^2$, and $\gamma := 2(\eta^2 \mathcal{M}^2 + \kappa^2)$,

$$\mathbb{E}[\|\mathbf{Z}_n\|^2] \leqslant \|\mathbf{Z}_n\|_{p,2}^2 \leqslant (\phi + C_p \gamma)^n \|\mathbf{Z}_0\|_{p,2}^2 \leqslant (1 + \eta \lambda_1)^{2n} \exp(C_p n \gamma) \|\mathbf{Z}_0\|_{p,2}^2.$$

Set $p := \max\left(2, \sqrt{\frac{2\log d}{n\gamma}}\right)$. Then, $\|\mathbf{Z}_0\|_{p,2} = d^{\frac{1}{p}} \leqslant \exp\left(\frac{pn\gamma}{2}\right)$. Therefore,

$$\mathbb{E}[\|\mathbf{Z}_n\|^2] \leqslant (1 + \eta \lambda_1)^{2n} \exp(2pn\gamma) = \exp\left(2\sqrt{2n\gamma \max\{2n\gamma, \log(d)\}}\right) (1 + \eta \lambda_1)^{2n}.$$

For the second result, set $p := 2\left(1 + \log\left(d\right)\right)$. Then, $C_p n \gamma \leqslant 1$ by assumption and $\|\mathbf{Z}_0\|_p = d^{1/p} \leqslant \sqrt{e}$. By Proposition A.1,

$$
\begin{aligned}
\mathbb{E}\left[\|\mathbf{Z}_n - \mathbb{E}\left[\mathbf{Z}_n\right]\|^2\right] &\leqslant \|\mathbf{Z}_n - \mathbb{E}\left[\mathbf{Z}_n\right]\|_{p,2}^2 \leqslant \left(\exp\left(C_p n \gamma\right) - 1\right)\left(1 + \eta\lambda_1\right)^n \|\mathbf{Z}_0\|_p^2 \\
&\leqslant e^2 C_p n \gamma \left(1 + \eta\lambda_1\right)^n \\
&< 2e^2 n \gamma \left(1 + 2\log\left(d\right)\right)\left(1 + \eta\lambda_1\right)^n.
\end{aligned}
$$

$\square$

## C.2 Proof of Lemma 4

**Lemma A.8** (Lemma 4 in main paper). *Let Assumption 1 hold and $\eta$ be set according to Lemma A.2 with $b = n$. Define $\gamma := 2(\eta^2 \mathcal{M}^2 + \kappa^2)$. If $2n\gamma\left(1 + 2\log\left(d\right)\right) \leqslant 1$, then*

$$
\mathbb{E}\left[\mathrm{Tr}\left(\mathbf{V}_\perp^\top \mathbf{Z}_n \mathbf{Z}_n^\top \mathbf{V}_\perp\right)\right] \leqslant \exp\left(2\eta n\lambda_1 + \eta^2 n\left(\mathcal{V}_0 + \lambda_1^2\right)\right)\left[\frac{d}{\exp\left(2\eta n\left(\lambda_1 - \lambda_2\right)\right)} + \frac{5\left(\eta^2\mathcal{V}_0 + \kappa_1\right)}{\eta\left(\lambda_1 - \lambda_2\right)}\right].
$$

*Proof.* Let $\beta_i := \mathbb{E}\left[\mathrm{Tr}\left(\mathbf{V}_\perp^\top \mathbf{Z}_i \mathbf{Z}_i^\top \mathbf{V}_\perp\right)\right]$ for all $0 \leqslant i \leqslant n$. Then, for $i \in [n]$,

$$
\begin{aligned}
\beta_i &= \mathbb{E}\left[\mathrm{Tr}\left(\mathbf{V}_\perp^\top \left(\mathbf{I} + \mathbf{A}_i\right)\mathbf{Z}_{i-1}\mathbf{Z}_{i-1}^\top \left(\mathbf{I} + \mathbf{A}_i^\top\right)\mathbf{V}_\perp\right)\right] \\
&= \mathbb{E}\left[\mathbb{E}\left[\mathrm{Tr}\left(\mathbf{V}_\perp^\top \left(\mathbf{I} + \mathbf{A}_i\right)\mathbf{Z}_{i-1}\mathbf{Z}_{i-1}^\top \left(\mathbf{I} + \mathbf{A}_i^\top\right)\mathbf{V}_\perp\right)|\mathcal{F}_{i-}\right]\right] \\
&= \mathbb{E}\left[\mathbb{E}\left[\mathrm{Tr}\left(\mathbf{V}_\perp^\top \left(\mathbf{I} + \eta\mathbf{Y}_i\right)\mathbf{Z}_{i-1}\mathbf{Z}_{i-1}^\top \left(\mathbf{I} + \eta\mathbf{Y}_i\right)\mathbf{V}_\perp\right)|\mathcal{F}_{i-}\right]\right] \\
&\quad + \mathbb{E}\left[\mathbb{E}\left[\mathrm{Tr}\left(\mathbf{V}_\perp^\top \mathbf{\Xi}_i \mathbf{Z}_{i-1}\mathbf{Z}_{i-1}^\top \mathbf{\Xi}_i^\top \mathbf{V}_\perp\right)|\mathcal{F}_{i-}\right]\right].
\end{aligned}
$$

The last line used $\mathbb{E}\left[\mathbf{\Xi}_i|\mathcal{F}_{i-}\right] = \mathbf{0}$ and that $\mathbf{Z}_{i-1}$ is measurable with respect to $\mathcal{F}_{i-}$. In other words,

$$
\beta_i = \mathbb{E}\left[\mathrm{Tr}\left(\mathbf{V}_\perp^\top \left(\mathbf{I} + \eta\mathbf{Y}_i\right)\mathbf{Z}_{i-1}\mathbf{Z}_{i-1}^\top \left(\mathbf{I} + \eta\mathbf{Y}_i\right)\mathbf{V}_\perp\right)\right] + \mathbb{E}\left[\mathrm{Tr}\left(\mathbf{Z}_{i-1}^\top \mathbb{E}\left[\mathbf{\Xi}_i^\top \mathbf{V}_\perp \mathbf{V}_\perp^\top \mathbf{\Xi}_i|\mathcal{F}_{i-}\right]\mathbf{Z}_{i-1}\right)\right].
$$

For the first term, following the analysis of Lemma 10 of [JJK+16],

$$
\begin{aligned}
\mathbb{E}\left[\mathrm{Tr}\left(\mathbf{V}_\perp^\top \left(\mathbf{I} + \eta\mathbf{Y}_i\right)\mathbf{Z}_{i-1}\mathbf{Z}_{i-1}^\top \left(\mathbf{I} + \eta\mathbf{Y}_i\right)\mathbf{V}_\perp\right)\right] &\leqslant \left(1 + 2\eta\lambda_2 + \eta^2\left(\mathcal{V}_0 + \lambda_1^2\right)\right)\beta_{i-1} + \eta^2\mathcal{V}_0\left\|\mathbb{E}\left[\mathbf{Z}_{i-1}\mathbf{Z}_{i-1}^\top\right]\right\|_2 \\
&\leqslant \left(1 + 2\eta\lambda_2 + \eta^2\left(\mathcal{V}_0 + \lambda_1^2\right)\right)\beta_{i-1} + \eta^2\mathcal{V}_0\mathbb{E}\left[\|\mathbf{Z}_{i-1}\|_2^2\right].
\end{aligned}
$$
(A.19)

The second term can be bounded as

$$
\begin{aligned}
\mathbb{E}\left[\mathrm{Tr}\left(\mathbf{Z}_{i-1}^\top \mathbb{E}\left[\mathbf{\Xi}_i^\top \mathbf{V}_\perp \mathbf{V}_\perp^\top \mathbf{\Xi}_i|\mathcal{F}_{i-}\right]\mathbf{Z}_{i-1}\right)\right] &= \mathbb{E}\left[\mathrm{Tr}\left(\mathbb{E}\left[\mathbf{\Xi}_i^\top \mathbf{V}_\perp \mathbf{V}_\perp^\top \mathbf{\Xi}_i|\mathcal{F}_{i-}\right]\mathbf{Z}_{i-1}\mathbf{Z}_{i-1}^\top\right)\right] \\
&\leqslant \mathbb{E}\left[\mathbb{E}\left[\mathrm{Tr}\left(\mathbf{\Xi}_i^\top \mathbf{V}_\perp \mathbf{V}_\perp^\top \mathbf{\Xi}_i\right)|\mathcal{F}_{i-}\right]\left\|\mathbf{Z}_{i-1}\mathbf{Z}_{i-1}^\top\right\|_2\right] \\
&\leqslant \kappa_1 \mathbb{E}\left[\left\|\mathbf{Z}_{i-1}\mathbf{Z}_{i-1}^\top\right\|_2\right].
\end{aligned}
$$
(A.20)

Combining (A.19) and (A.20), we obtain the recurrence

$$
\beta_i \leqslant \left(1 + 2\eta\lambda_2 + \eta^2\left(\mathcal{V}_0 + \lambda_1^2\right)\right)\beta_{i-1} + \left(\eta^2\mathcal{V}_0 + \kappa_1\right)\mathbb{E}[\|\mathbf{Z}_{i-1}\|_2^2].
$$

By Lemma A.7, we have for $\gamma := 2(\eta^2\mathcal{M}^2 + \kappa^2)$,

$$
\begin{aligned}
\beta_i &\leqslant \left(1 + 2\eta\lambda_2 + \eta^2\left(\mathcal{V}_0 + \lambda_1^2\right)\right)\beta_{i-1} + \left(\eta^2\mathcal{V}_0 + \kappa_1\right)\exp\left(2\sqrt{2n\gamma\log d}\right)\left(1 + \eta\lambda_1\right)^{2(i-1)} \\
&\leqslant \exp\left(2\eta\lambda_2 + \eta^2\left(\mathcal{V}_0 + \lambda_1^2\right)\right)\beta_{i-1} + s\exp\left(2\eta\lambda_1 + \eta^2(\mathcal{V}_0 + \lambda_1^2)\right)^{i-1},
\end{aligned}
$$

where $s = (\eta^2 \mathcal{V}_0 + \kappa_1) \exp\left(2\sqrt{2n\gamma \log d}\right)$. Unrolling the recursion,

$$\beta_n \leqslant \exp\left(2\eta n\lambda_1 + \eta^2 n \left(\mathcal{V}_0 + \lambda_1^2\right)\right) \left[\exp\left(-2\eta n\left(\lambda_1 - \lambda_2\right)\right)\beta_0 + s. \sum_{t=0}^{n-1} \left(\frac{\exp\left(2\eta\lambda_2 + \eta^2\left(\mathcal{V}_0 + \lambda_1^2\right)\right)}{\exp\left(2\eta\lambda_1 + \eta^2\left(\mathcal{V}_0 + \lambda_1^2\right)\right)}\right)^{2(n-1-t)}\right]$$

$$\leqslant \exp\left(2\eta n\lambda_1 + \eta^2 n \left(\mathcal{V}_0 + \lambda_1^2\right)\right) \left[\exp\left(-2\eta n\left(\lambda_1 - \lambda_2\right)\right)\beta_0 + \frac{s}{1 - \exp(-2\eta\left(\lambda_1 - \lambda_2\right))}\right]$$

$$\leqslant \exp\left(2\eta n\lambda_1 + \eta^2 n \left(\mathcal{V}_0 + \lambda_1^2\right)\right) \left[\exp\left(-2\eta n\left(\lambda_1 - \lambda_2\right)\right)\beta_0 + \frac{2.35s}{2\eta\left(\lambda_1 - \lambda_2\right)}\right]$$

$$\leqslant \exp\left(2\eta n\lambda_1 + \eta^2 n \left(\mathcal{V}_0 + \lambda_1^2\right)\right) \left[\exp\left(-2\eta n\left(\lambda_1 - \lambda_2\right)\right)d + \frac{5(\eta^2\mathcal{V}_0 + \kappa_1)}{\eta\left(\lambda_1 - \lambda_2\right)}\right]$$

where the third inequality holds because $x \leqslant 2.35(1 - e^{-x})$ for $x \leqslant 2$ and the last inequality holds because $\beta_0 \leqslant d$ and $\frac{2.35 \exp\left(2\sqrt{2n\gamma \log d}\right)}{2} \leqslant \frac{2.35 \exp(\sqrt{2})}{2} < 5$. $\qquad\square$

# D    Proofs of Theorems 1, 2, and 3

### D.1    Proof of Theorem 1

We are now ready to present the proof of Theorem 1, which follows from the following Theorem A.4 and setting a constant failure probability for $\theta$.

**Theorem A.4.** *Fix $\theta \in (0, 1)$. Then, for $\mathbf{w}$ being the output of Algorithm 1, under assumption 1, learning rate $\eta = \frac{\alpha \log n}{b(\lambda_1 - \lambda_2)}$ with $\alpha$ is set as in Lemma A.2, $\kappa_1 \leqslant 1/2$, and*

$$\sqrt{2e^2 b\gamma \log\left(d/\theta\right)} \leqslant \frac{1}{2},$$

*where $\gamma := 2(\eta^2\mathcal{M}^2 + \kappa^2)$. Then, with probability at least $1 - 3\theta$,*

$$\sin^2(\mathbf{w}, \mathbf{v}_1) \leqslant \frac{24 \log\left(1/\theta\right)}{\theta^3} \left[\frac{d}{\exp\left(2\alpha\log(n)\right)} + \frac{5\left(\eta^2\mathcal{V}_0 + \kappa_1\right)}{\eta\left(\lambda_1 - \lambda_2\right)}\right] + 8\kappa_1.$$

*Proof.* Note that by Algorithm 1 and the definition of $\mathbf{Z}$ in (A.18),

$$\mathbf{u}_b = \frac{\mathbf{Z}_b \mathbf{u}_0}{\|\mathbf{Z}_b \mathbf{u}_0\|}.$$

Since $\mathbf{v}_1 \mathbf{v}_1^\top + \mathbf{V}_\perp \mathbf{V}_\perp^\top = \mathbf{I}_d$,

$$\sin^2\left(\mathbf{u}_b, \mathbf{v}_1\right) = 1 - \left(\mathbf{u}_b^\top \mathbf{v}_1\right)^2 = \left\|\frac{\mathbf{V}_\perp \mathbf{V}_\perp^\top \mathbf{Z}_b \mathbf{u}_0}{\|\mathbf{Z}_b \mathbf{u}_0\|}\right\|^2.$$

By Lemma 6 from [JJK$^+$16], with probability at least $1 - \theta$,

$$\sin^2\left(\mathbf{u}_b, \mathbf{v}_1\right) \leqslant \frac{2.5 \log\left(1/\theta\right)}{\theta^2} \frac{\mathrm{Tr}\left(\mathbf{V}_\perp^\top \mathbf{Z}_b \mathbf{Z}_b^\top \mathbf{V}_\perp\right)}{\mathbf{v}_1^\top \mathbf{Z}_b \mathbf{Z}_b^\top \mathbf{v}_1}.$$

By Lemma A.7 with $q = 2$ and $p = 2\left(1 + \log\left(d\right)\right)$,

$$\mathbb{E}\left[\|\mathbf{Z}_b - (\mathbf{I} + \eta\mathbf{\Sigma})^b\|\right] \leqslant \|\mathbf{Z}_b - (\mathbf{I} + \eta\mathbf{\Sigma})^b\|_{p,2} \leqslant \sqrt{e^2 b\gamma \left(1 + 2\log\left(d\right)\right)}\left(1 + \eta\lambda_1\right)^b. \quad \text{(A.21)}$$

For the numerator, we use Lemma A.8 and Markov's inequality to get

$$\mathrm{Tr}\left(\mathbf{V}_\perp^\top \mathbf{Z}_b \mathbf{Z}_b^\top \mathbf{V}_\perp\right) \leqslant \frac{1}{\theta}\exp\left(2\eta b\lambda_1 + \eta^2 b \left(\mathcal{V}_0 + \lambda_1^2\right)\right) \left[\frac{d}{\exp\left(2\eta b\left(\lambda_1 - \lambda_2\right)\right)} + \frac{5\left(\eta^2\mathcal{V}_0 + \kappa_1\right)}{\eta\left(\lambda_1 - \lambda_2\right)}\right].$$
$$\text{(A.22)}$$

with probability at least $1 - \theta$.

The denominator can be bounded as

$$\left\| \mathbf{Z}_b^\top \mathbf{v}_1 \right\| \geqslant \left\| (\mathbf{I} + \eta \boldsymbol{\Sigma})^b \, \mathbf{v}_1 \right\| - \left\| \left( \mathbf{Z}_b - (\mathbf{I} + \eta \boldsymbol{\Sigma})^b \right)^\top \mathbf{v}_1 \right\| \geqslant (1 + \eta \lambda_1)^b - \left\| \mathbf{Z}_b - (\mathbf{I} + \eta \boldsymbol{\Sigma})^b \right\|.$$

Using Lemma A.6, with probability atleast $1 - \theta$,

$$
\begin{aligned}
\|\mathbf{Z}_b \mathbf{v}_1\| &\geqslant (1 + \eta \lambda_1)^b - \sqrt{2 e^2 b \gamma \log (d/\theta)} \, (1 + \eta \lambda_1)^b \\
&= (1 + \eta \lambda_1)^b \left( 1 - \sqrt{2 e^2 b \gamma \log (d/\theta)} \right) \\
&\geqslant \exp \left( \eta \lambda_1 b - \eta^2 \lambda_1^2 b \right) \left( 1 - \sqrt{2 e^2 b \gamma \log (d/\theta)} \right).
\end{aligned}
\tag{A.23}
$$

where the last line follows since $(1 + x) \geqslant \exp \left( x - x^2 \right)$ for all $x \geqslant 0$. From equations (A.22), (A.23), and the assumption $\sqrt{2 e^2 b \gamma \log (d/\theta)} \leqslant 1/2$, it follows that with probability $1 - 3\theta$,

$$\sin^2 \left( \mathbf{u}_b, \mathbf{v}_1 \right) \leqslant \frac{12 \log (1/\theta)}{\theta^3} \left[ \frac{d}{\exp (2\alpha \log(n))} + \frac{5 \left( \eta^2 \mathcal{V}_0 + \kappa_1 \right)}{\eta \left( \lambda_1 - \lambda_2 \right)} \right]. \tag{A.24}$$

Since $\mathbf{w} \leftarrow \mathbf{Q}(\mathbf{u}_b, \mathcal{Q})$, by Lemma A.9 and using $\|\boldsymbol{\xi}\| \leqslant \kappa \leqslant 0.5$,

$$\sin^2 \left( \mathbf{w}, \mathbf{u}_b \right) \leqslant \frac{\|\boldsymbol{\xi}\|^2}{\|\mathbf{u}_b + \boldsymbol{\xi}\|^2} \leqslant \frac{\|\boldsymbol{\xi}\|^2}{(\|\mathbf{u}_b\| - \|\boldsymbol{\xi}\|)^2} \leqslant \frac{\kappa^2}{0.5^2} \leqslant 4\kappa^2. \tag{A.25}$$

The result follows by using equations (A.24), (A.25), and Lemma A.5. $\qquad\square$

## D.2 Proofs of Theorems 2 and 3

Next, we apply Theorem A.4 to analyze the quantized version of Oja's algorithm as described in Algorithm 1. The idea is to show that the error from the rounding operation can be incorporated into the noise in the iterates of Oja's algorithm, which have mean zero. For this subsection, we will use:

$$\mathbf{D}_i = \sum_{j \in B_i} \frac{\mathbf{X}_j \mathbf{X}_j^T}{n/b},$$

where $\mathbf{A}_i = \eta \left( \mathbf{D}_i + \boldsymbol{\xi}_{a,i} \mathbf{u}_{i-1}^T \right) + \boldsymbol{\xi}_{2,i} \mathbf{u}_{i-1}^T + (\mathbf{I} + \eta \mathbf{D}_i) \boldsymbol{\xi}_{1,i} \mathbf{u}_{i-1}^T$.

We first state and prove some intermediate results needed to prove Theorems 2 and Theorems 3.

**Theorem A.5.** *Let* $d, n, b \in \mathbb{N}$, *and let* $\{\mathbf{X}_i\}_{i \in [n]}$ *be a set of $n$ IID vectors in $\mathbb{R}^d$ satisfying assumption 1. Let* $\eta := \frac{\alpha \log n}{b(\lambda_1 - \lambda_2)}$ *be the learning rate set as in Lemma A.2. Suppose the quantization grid* $\mathcal{Q} = \mathcal{Q}_L$, *and* $\sqrt{4 e^2 b (4\eta^2 + 9\delta^2 d) \log (d/\theta)} \leqslant \frac{1}{2}$. *Then, with probability at least $0.9$, the output $\mathbf{w}$ of Algorithm 1 satisfies*

$$\sin^2(\mathbf{w}, \mathbf{v}_1) \leqslant \frac{24 \log (1/\theta)}{\theta^3} \left[ \frac{d}{n^{2\alpha}} + \frac{5\alpha \mathcal{V} \log n}{n \left( \lambda_1 - \lambda_2 \right)^2} + \frac{30 b \delta^2 d}{\alpha \log n} \right] + 48 \delta^2 d.$$

*Proof.* In order to apply Theorem 1, we come up with valid choices of $\mathcal{V}_0$, $\kappa$, and $\kappa_1$.

Since each $\mathbf{D}_i$ is symmetric and $\{\mathbf{X}_i\}_{i \in [n]}$ are independent,

$$\left\| \mathbb{E}[(\mathbf{D}_i - \boldsymbol{\Sigma})(\mathbf{D}_i - \boldsymbol{\Sigma})^T] \right\| = \left\| \frac{1}{n/b} \mathbb{E}[(\mathbf{X}_1 \mathbf{X}_1^T - \boldsymbol{\Sigma})^2] \right\| \leqslant \frac{b\mathcal{V}}{n} =: \mathcal{V}_0. \tag{A.26}$$

Next,

$$\boldsymbol{\Xi}_i = \eta \boldsymbol{\xi}_{a,i} \mathbf{u}_{i-1}^T + \boldsymbol{\xi}_{2,i} \mathbf{u}_{i-1}^T + (\mathbf{I} + \eta \mathbf{D}_i) \boldsymbol{\xi}_{1,i} \mathbf{u}_{i-1}^T.$$

Also observe that

$$\mathbb{E}[\boldsymbol{\xi}_{1,i}|\mathcal{F}_{i-}] = 0, \qquad \mathbb{E}[\boldsymbol{\xi}_{a,i}|\mathcal{F}_{i-}] = 0, \qquad \mathbb{E}[\boldsymbol{\xi}_{2,i}|\boldsymbol{\xi}_{a,i}, \boldsymbol{\xi}_{1,i}, \mathcal{F}_{i-}] = 0, \tag{A.27}$$

By equation A.27,

$$\mathbb{E}[\boldsymbol{\Xi}_i^T\boldsymbol{\Xi}_i|\mathcal{F}_{i-}] = \mathbb{E}[\eta^2\mathbf{u}_{i-1}\boldsymbol{\xi}_{a,i}^T\boldsymbol{\xi}_{a,i}\mathbf{u}_{i-1}^T + \mathbf{u}_{i-1}\boldsymbol{\xi}_{2,i}^T\boldsymbol{\xi}_{2,i}\mathbf{u}_{i-1}^T + \mathbf{u}_{i-1}\boldsymbol{\xi}_{1,i}^T(I+\eta\mathbf{D}_i)(I+\eta\mathbf{D}_i)^T\boldsymbol{\xi}_{2,i}\mathbf{u}_{i-1}^T|\mathcal{F}_{i-}]$$

$$\implies \left\|\mathbb{E}[\boldsymbol{\Xi}_i^T\boldsymbol{\Xi}_i|\mathcal{F}_{i-}]\right\|_F \leqslant \eta^2\delta^2 d + \delta^2 d + (1+\eta)^2\delta^2 d \leqslant 6\delta^2 d =: \kappa_1.$$

As for $\kappa$, we have

$$\|\boldsymbol{\Xi}_i\| \leqslant 2(1+\eta)\delta\sqrt{d} \leqslant 3\delta\sqrt{d} =: \kappa$$

We are now ready to obtain the sin-squared error. Note that $\mathcal{M} \leqslant 2$, since $\|\mathbf{X}_i\| \leqslant 1$ almost surely, for all $i \in [n]$. By Theorem A.4, with probability at least $1 - 3\theta$,

$$\sin^2(\mathbf{w}, \mathbf{v}_1) \leqslant \frac{24\log(1/\theta)}{\theta^3}\left[\frac{d}{\exp(2\alpha\log(n))} + \frac{5\left(\eta^2\mathcal{V}_0 + \kappa_1\right)}{\eta(\lambda_1 - \lambda_2)}\right] + 8\kappa_1.$$

as long as $\sqrt{2e^2 b\gamma\log(d/\theta)} \leqslant \frac{1}{2}$. Our parameter choices are $\mathcal{V}_0 = \frac{b\mathcal{V}}{n}, \kappa = 3\delta\sqrt{d}$, and $\kappa_1 = 6\delta^2 d$.

$$\sin^2(\mathbf{w}, \mathbf{v}_1) \leqslant \frac{24\log(1/\theta)}{\theta^3}\left[\frac{d}{n^{2\alpha}} + \frac{5\alpha\mathcal{V}\log n}{n(\lambda_1 - \lambda_2)^2} + \frac{30b\delta^2 d}{\alpha\log n}\right] + 48\delta^2 d.$$

$\square$

**Lemma A.9.** *Let* $\mathbf{u} = Q(\mathbf{w}, \mathcal{Q}_{NL})$, *where* $\mathbf{u} \in \mathbb{R}^d$ *and* $\mathcal{Q}_{NL}$ *is defined in equation 4. Then,*

$$\|\mathbf{w} - Q(\mathbf{w}, \mathcal{Q}_{NL})\| \leqslant \delta_0\sqrt{d} + \|\mathbf{w}\|\zeta$$

*Proof.* Let $\boldsymbol{\xi} = Q(\mathbf{w}, \mathcal{Q}_{NL}) - \mathbf{w}$. Say $\mathbf{w}_i > 0$. Let $k$ be the unique integer such that $\mathbf{w}_i \in [q_k, q_{k+1}]$. Equivalently for negative $\mathbf{w}_i$, say the bin is $[-q_{k+1}, -q_k]$. We have:

$$|\boldsymbol{\xi}_i| \leqslant q_{k+1} - q_k \leqslant \delta_0 + \zeta q_k \leqslant |\mathbf{w}_i|\zeta + \delta_0$$

Thus we have:

$$\|\boldsymbol{\xi}\| \leqslant \delta_0\sqrt{d} + \|\mathbf{w}\|\zeta.$$

$\square$

**Theorem A.6.** *Fix* $\theta \in (0, 1)$. *Let the initial vector* $\mathbf{u}_0 \sim \mathcal{N}(0, \mathbf{I})$. *Let the number of batches* $b$ *and quantization scale* $\delta$ *be such that* $\sqrt{4e^2 b(4\eta^2 + 32\delta_0^2 d + 98\zeta^2)\log(d/\theta)} \leqslant 1/2$. *Then, under assumption 1with* $\eta$ *set as* $\frac{\alpha\log n}{b(\lambda_1 - \lambda_2)}$, *where* $\alpha$ *is set as in Lemma A.2,* $\delta_0\sqrt{d} \leqslant 0.25$, *and* $\zeta \leqslant 0.25$, *with probability at least* $1 - 3\theta$, *the output* $\mathbf{w}_b$ *of Algorithm 1 gives:*

$$\sin^2(\mathbf{w}, \mathbf{v}_1) \leqslant \frac{24\log(1/\theta)}{\theta^3}\left[\frac{d}{n^{2\alpha}} + \frac{5\alpha\mathcal{V}\log n}{n(\lambda_1 - \lambda_2)^2} + \frac{5b(4\delta_0\sqrt{d} + 7\zeta)^2}{\alpha\log n}\right] + 8(4\delta_0\sqrt{d} + 7\zeta)^2.$$

*Proof.* In order to apply Theorem 1 we need to bound $\mathcal{V}$, $\kappa$ and $\kappa_1$. We start with the first. For us, $\mathbf{D}_i$ is defined in Eq 9. Let $\mathcal{R}_i$ denote the random variables in the quantization up to and including the $i^{th}$ update.

Our analysis is analogous to the previous theorem. Note that the $\mathcal{V}_0$ parameter is as in Eq A.26.

Now we will work out $\kappa$ and $\kappa_1$ since those are the only quantities that change for the nonlinear quantization. Recall that we have,

$$\boldsymbol{\Xi}_i = \eta\boldsymbol{\xi}_{a,i}\mathbf{u}_{i-1}^T + \boldsymbol{\xi}_{2,i}\mathbf{u}_{i-1}^T + (\mathbf{I} + \eta\mathbf{D}_i)\boldsymbol{\xi}_{1,i}\mathbf{u}_{i-1}^T.$$

We have,

$$\mathbb{E}[\boldsymbol{\Xi}_i^T\boldsymbol{\Xi}_i|\mathcal{F}_{i-}]$$
$$= \eta^2\mathbb{E}[\mathbf{u}_{i-1}\boldsymbol{\xi}_{a,i}^T\boldsymbol{\xi}_{a,i}\mathbf{u}_{i-1}^T|\mathcal{F}_{i-}] + \mathbb{E}[\mathbf{u}_{i-1}\boldsymbol{\xi}_{2,i}^T\boldsymbol{\xi}_{2,i}\mathbf{u}_{i-1}^T|\mathcal{F}_{i-}] + \mathbb{E}[\mathbf{u}_{i-1}\boldsymbol{\xi}_{1,i}^T(I+\eta\mathbf{D}_i)(I+\eta\mathbf{D}_i)^T\boldsymbol{\xi}_{2,i}\mathbf{u}_{i-1}^T|\mathcal{F}_{i-}]$$

Now we obtain the Frobenius norm of $\boldsymbol{\xi}_{a,i}$, $\boldsymbol{\xi}_1$, and $\boldsymbol{\xi}_2$ under the nonlinear quantization. We start with the norm of $\mathbf{w}_i$, a quantized version of a unit vector $\mathbf{u}_{i-1}$.

By Lemma A.9, $\|\mathbf{w}_i\| \leqslant 1 + \delta_0\sqrt{d} + \zeta$. Let $\mathbf{s}_j = \mathbf{X}_j(\mathbf{X}_j^T\mathbf{w}_i)$. Then,

$$\|\mathbf{s}_j\| \leqslant \|\mathbf{w}_i\| \leqslant 1 + \delta_0\sqrt{d} + \zeta.$$

Another application of Lemma A.9 gives:

$$\|\boldsymbol{\xi}_{a,j,i}\| = \|\mathsf{Q}(\mathbf{s}_j, \mathcal{Q}_{NL}) - \mathbf{s}_j\| \leqslant \delta_0\sqrt{d} + (1 + \delta_0\sqrt{d} + \zeta)\zeta \leqslant \delta_0\sqrt{d} + 1.5\zeta$$

which implies $\|\boldsymbol{\xi}_{a,i}\| \leqslant \delta_0\sqrt{d} + 1.5\zeta$. Next, we bound $\boldsymbol{\xi}_{1,i} = \mathsf{Q}(\mathbf{u}_{i-1}, \mathcal{Q}_{NL}) - \mathbf{u}_{i-1}$. By Lemma A.9,

$$\|\boldsymbol{\xi}_{1,i}\| \leqslant \delta_0\sqrt{d} + \zeta\,\|\mathbf{u}_{i-1}\| = \delta_0\sqrt{d} + \zeta.$$

Finally we bound $\boldsymbol{\xi}_{2,i}$. Recall that:

$$\mathbf{y}_i = \frac{\sum_{j\in\mathcal{B}_j}\mathbf{X}_j(\mathbf{X}_j^T\mathbf{w}_i)}{n/b} + \boldsymbol{\xi}_{a,i}$$

$$\boldsymbol{\xi}_{2,i} = \mathsf{Q}\left(\mathbf{y}_i, \delta\right) - \mathbf{y}_i$$

Since each $\left\|\mathbf{X}_j\mathbf{X}_j^\top\mathbf{w}_i\right\| \leqslant 1 + \delta_0\sqrt{d} + \zeta$,

$$\|\mathbf{y}_i\| \leqslant 1 + \delta_0\sqrt{d} + \zeta + \|\boldsymbol{\xi}_{a,i}\| \leqslant 1 + 2\delta_0\sqrt{d} + 2.5\zeta \leqslant 3.25.$$

By Lemma A.9,

$$\|\boldsymbol{\xi}_{2,i}\| \leqslant \delta_0\sqrt{d} + \zeta\|\mathbf{y}_i\| \leqslant \delta_0\sqrt{d} + 3.25\zeta.$$

In all, it follows that

$$\|\boldsymbol{\Xi}_i\| \leqslant \eta\|\boldsymbol{\xi}_{a,i}\| + \|\boldsymbol{\xi}_{2,i}\| + (1+\eta)\|\boldsymbol{\xi}_{1,i}\| \leqslant (\delta_0\sqrt{d} + 1.5\zeta) + (\delta_0\sqrt{d} + 3.25\zeta) + 2(\delta_0\sqrt{d} + \zeta) \leqslant 4\delta_0\sqrt{d} + 7\zeta =: \kappa.$$

We are ready to obtain the sin-squared error. Note that $\mathcal{M} \leqslant 2$, since $\|\mathbf{X}_i\| \leqslant 1$ almost surely, for all $i \in [n]$. By Theorem A.4, with probability at least $1 - 3\theta$,

$$\sin^2(\mathbf{w}, \mathbf{v}_1) \leqslant \frac{24\log\left(1/\theta\right)}{\theta^3}\left[\frac{d}{\exp\left(2\alpha\log(n)\right)} + \frac{5\left(\eta^2\mathcal{V}_0 + \kappa_1\right)}{\eta\left(\lambda_1 - \lambda_2\right)}\right] + 8\kappa_1.$$

as long as $\sqrt{2e^2b\gamma\log\left(d/\theta\right)} \leqslant \frac{1}{2}$. Our parameter choices are $\mathcal{V}_0 = \frac{b\mathcal{V}}{n}, \kappa = 4\delta_0\sqrt{d} + 7\zeta$, and $\kappa_1 = (4\delta_0\sqrt{d} + 7\zeta)^2$. Therefore,

$$\sin^2(\mathbf{w}, \mathbf{v}_1) \leqslant \frac{24\log\left(1/\theta\right)}{\theta^3}\left[\frac{d}{n^{2\alpha}} + \frac{5\alpha\mathcal{V}\log n}{n\left(\lambda_1 - \lambda_2\right)^2} + \frac{5b(4\delta_0\sqrt{d} + 7\zeta)^2}{\alpha\log n}\right] + 8(4\delta_0\sqrt{d} + 7\zeta)^2.$$

$\square$

### D.2.1 Finishing the Proofs of Theorems 2 and 3

*Proof of Theorem 2.* For the linear quantization scheme, we apply Theorem A.5 with $\theta = 1/30$ and $b = \Theta\left(\frac{\alpha^2\log^2 n\log d}{(\lambda_1-\lambda_2)^2}\right)$. Moreover, since $\delta = \tilde{O}\left(\frac{\lambda_1-\lambda_2}{\alpha\sqrt{d}}\right)$, the condition $\sqrt{4e^2b(4\eta^2 + 9\delta^2d)\log\left(d/\theta\right)} \leqslant \frac{1}{2}$ holds. The Theorem follows by substituting these values into the bound of Theorem A.5.

The proof of the logarithmic scheme follows analogously from Theorem A.6. $\square$

*Proof of Theorem 3.* We set $\theta = 1/30$. For the linear quantization scheme, we apply Theorem A.5 with $b = n$. Moreover, since $\delta = 2^{2-\beta} = O\left(\min\left(\frac{\lambda_1-\lambda_2}{\alpha\sqrt{d}\log(n)}, \frac{1}{\sqrt{dn}}\right)\right)$, the condition $\sqrt{4e^2b(4\eta^2 + 9\delta^2d)\log\left(d/\theta\right)} \leqslant \frac{1}{2}$ holds. The Theorem follows by substituting these values into the bound of Theorem A.5.

For the non-linear scheme, the proof follows analogously from Theorem A.6. $\square$

### D.3 Optimal Choice of Parameters

We want to minimize the quantity

$$\kappa_1 := \zeta^2 + \delta_0^2 d,$$

where $\zeta = 2^{-\beta_m}$ and $\delta_0 = 4 \cdot 2^{-2^{\beta_e - 1}}$. Here, $\beta_m$ and $\beta_e$ are the number of bits used by the mantissa and the exponent, respectively, and satisfy the constraint

$$\beta_m + \beta_e = \beta.$$

Then,

$$\zeta^2 + \delta_0^2 d = 2^{-2(\beta - \beta_e)} + 16d2^{-2^{\beta_e}} =: f(\beta_e).$$

To find $\beta_e$ that minimizes $f(\beta_e)$ we differentiate with respect to $\beta_e$ and set it to 0.

$$f'(\beta_e) = 2^{-2(\beta - \beta_e)} \cdot 2\ln 2 + 16d \cdot (2^{-2^{\beta_e}} \ln 2) \cdot (-2^{\beta_e} \ln 2)$$

$$= \left( \frac{2^{\beta_e}}{4^\beta} - 8d2^{-2^{\beta_e}} \ln 2 \right) 2^{\beta_e} \cdot 2\ln 2.$$

It is optimal to take $\beta_e$ such that

$$2^{\beta_e} 2^{2^{\beta_e}} = 8d \cdot 4^\beta \ln 2.$$

Equivalently, $\beta_e + 2^{\beta_e} = 2\beta + \log_2(8d\ln 2)$. This in particular implies

$$2^{\beta_e} < 2\beta + \log_2(8d\ln 2) < 2^{\beta_e + 1},$$

so

$$2\beta + \log_2(8d\ln 2) - 1 < \beta_e < \log_2\left(2\beta + \log_2(8d\ln 2)\right).$$

Therefore, we choose

$$\beta_e^* = \lceil \log_2\left(2\beta + \log_2(8d\ln 2)\right) \rceil, \quad \beta_m^* = \beta - \beta_e^*.$$

This choice of $\beta_e^*$ is valid as long as it does not make $\beta_m^*$ non-positive. This is true as long as $\beta \geqslant \max(8, \log_2(d))$. With these values of $\beta_e^*$ and $\beta_m^*$,

$$\zeta = 2^{\beta_e^* - \beta} < \frac{2^{(1 + \log_2(2\beta + \log_2(8d\ln 2)))}}{2^\beta} = \frac{2(2\beta + \log_2(8d\ln 2))}{2^\beta}$$

and

$$\delta_0^2 = \left(4 \cdot 2^{-2^{\beta_e^* - 1}}\right)^2 = 16 \cdot 2^{-2^{\beta_e^*}} \leqslant 16 \cdot 2^{-(2\beta + \log_2(8d\ln 2))} = \frac{2}{4^\beta d\ln 2}.$$

## E  Proof of Boosting Lemma (Lemma 3)

In this section, we present the proof of the boosting procedure. Our boosting procedure requires a modest assumption that the number of bits $\beta \geqslant 4$, which is already assumed in Section 3.4 while optimizing the parameters.

**Proof of Lemma 3**

*Proof.* For each $i \in [r]$, define the indicator random variable

$$\chi_i := \mathbb{1}\left(\sin^2(\mathbf{u}_i, \mathbf{v}) \leqslant \epsilon\right).$$

Then, by the guarantees of $\mathcal{A}$, $\Pr(\chi_i = 1) \geqslant 1 - p$, where $p = 0.1$. Let $\mathcal{S} := \{i \in [r] : \chi_i = 1\}$, and define the event

$$\mathcal{E} := \{|\mathcal{S}| > 0.6r\}.$$

The Chernoff bound for the sum of independent Bernoulli random variables gives

$$\mathbb{P}\left(|\mathcal{S}| \leqslant (1 - \theta)\,\mathbb{E}\left[|\mathcal{S}|\right]\right) \leqslant \exp\left(-\frac{\theta^2 \mathbb{E}\left[|\mathcal{S}|\right]}{2}\right) \quad \forall\,\theta \in (0, 1).$$

By linearity of expectation, $\mathbb{E}\left[|\mathcal{S}|\right] \geqslant (1 - p)r$. Setting $\theta = 1/3$,

$$\mathbb{P}\left(\mathcal{E}^c\right) \leqslant \mathbb{P}\left(|\mathcal{S}| \leqslant 0.6r\right) \leqslant e^{-r/20} \leqslant \delta.$$

It suffices to show that if the event $\mathcal{E}$ holds, then $\bar{\mathbf{u}}$ is well-defined and has small sin-squared error with $\mathbf{v}$. Recall,

$$\bar{\mathbf{u}} := \mathbf{u}_i \text{ such that } |\{j \in [r] : \tilde{\rho}\,(\mathbf{u}_i, \mathbf{u}_j) \leqslant 5\epsilon\}| \geqslant 0.5r,$$

Conditioned on $\mathcal{E}$, any $i$ that belongs to the set $\mathcal{S}$ satisfies $c_i \geqslant 0.6r$. Indeed, Lemma A.5 gives for any $i, j \in \mathcal{S}$

$$\sin^2(\mathbf{u}_i, \mathbf{u}_j) \leqslant 2\sin^2(\mathbf{u}_i, \mathbf{v}) + 2\sin^2(\mathbf{v}, \mathbf{u}_j) \leqslant 4\epsilon,$$

which implies

$$|\tilde{\rho}\,(\mathbf{u}_i, \mathbf{u}_j)| \leqslant \sin^2(\mathbf{u}_i, \mathbf{u}_j) + \epsilon \leqslant 5\epsilon$$

because $4\epsilon$ is within the range of the bounded grid $\mathcal{Q}_L(\epsilon) := \mathcal{Q}_L(\epsilon, \beta)$ defined in (11). Therefore, the algorithm does not return $\perp$ and $\bar{\mathbf{u}}$ is well-defined.

Now, $|\tilde{\rho}(\bar{\mathbf{u}}, \mathbf{u}_j)| \leqslant 5\epsilon$ for at least $0.5r$ indices $j \in [r]$ and $|\mathcal{S}| \geqslant 0.6r$. In particular, there exists an index $j^* \in \mathcal{S}$ for which $|\tilde{\rho}(\bar{\mathbf{u}}, \mathbf{u}_{j*})| \leqslant 5\epsilon$. Since $5\epsilon$ is strictly inside the grid $\mathcal{Q}_L(\epsilon)$, we get $\sin^2(\bar{\mathbf{u}}, \mathbf{u}_{j*}) \leqslant 6\epsilon$. We conclude

$$\sin^2(\bar{\mathbf{u}}, \mathbf{v}) \leqslant 2\sin^2(\bar{\mathbf{u}}, \mathbf{u}_j) + 2\sin^2(\mathbf{u}_j, \mathbf{v}) \leqslant 2(6\epsilon) + 2\epsilon = 14\epsilon.$$

$\square$

Theorem A.7 puts everything together and applies Lemma 3 to obtain the final high probability result.
**Theorem A.7.** *Suppose $\mathcal{A}$ is the Oja's algorithm with the setting of Theorem 2 or 3. Let $\epsilon$ be the probability $0.9$ error bound guaranteed by Theorem 2, $r = \lceil 20\log(1/\theta)\rceil$, and $m = nr$. Let $\{\mathbf{X}_i\}_{i\in[m]}$ be $n$ IID data drawn from a distribution satisfying assumption 1, and $\mathbf{u}_j \leftarrow \mathcal{A}(\{\mathbf{X}_i\}_{(j-1)n+1\leqslant i\leqslant jn}$ for all $j \in [r]$. Then, the output of algorithm 2 satisfies*

$$\sin^2(\bar{\mathbf{u}}, \mathbf{v}_1) \leqslant 14\epsilon$$

*with probability at least $1 - \theta$.*

*Proof.* The vectors $\mathbf{u}_1, \ldots, \mathbf{u}_r$ are mutually independent. By Theorem 2, $\Pr\left(\sin^2(\mathbf{u}_i, \mathbf{v}_1) > \epsilon\right) \leqslant 0.1 \,\forall\, i \in [r]$. Therefore, Lemma 3 applies and the theorem follows. $\square$

## F  Experimental Details

### F.1  Additional Synthetic Experiments

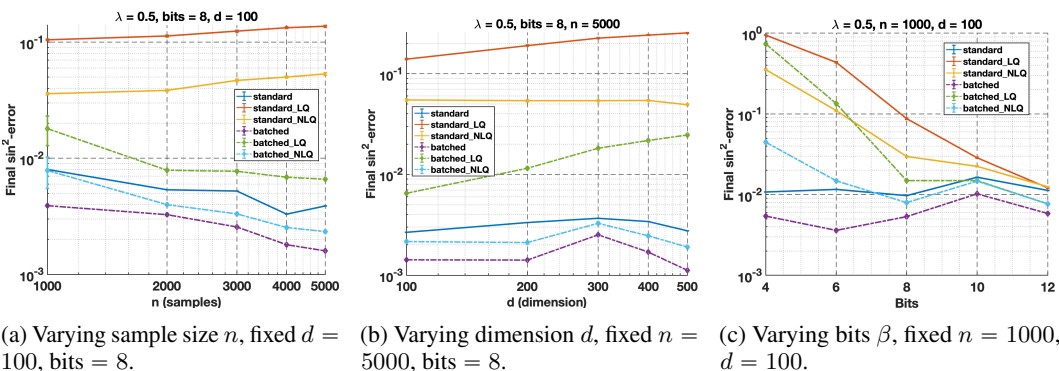

(a) Varying sample size $n$, fixed $d = 100$, bits $= 8$.

(b) Varying dimension $d$, fixed $n = 5000$, bits $= 8$.

(c) Varying bits $\beta$, fixed $n = 1000$, $d = 100$.

Figure A.1: Variation of $\sin^2$-error with: (a) sample size, (b) dimension, and (c) quantization bits.

We generate synthetic datasets via the procedure described in [LSW21]. The generation process takes as input the number of samples, $n$, the dimension $d$ and an eigenvalue decay parameter $\lambda$. We defer the details of the generation process to the Appendix Section F. Given the sample size $n$, dimension $d$, and decay exponent $\lambda$ in the eigenvalues, we first draw an $n \times d$ matrix $Z$ with independent entries uniformly distributed on $[-\sqrt{3}, \sqrt{3}]$ so that each coordinate has unit variance. We then build a kernel matrix $K \in \mathbb{R}^{d\times d}$ with entries $K_{ij} = \exp\left(-|i - j|^{0.01}\right)$ and define a variance profile $\sigma_i = 5\,i^{-\lambda}$

for $i = 1, \ldots, d$. The population covariance is formed as $\Sigma = (\sigma\sigma^\top) \circ K$, where $\circ$ denotes the Hadamard product. Computing the eigendecomposition of $\Sigma$ yields its square root $\Sigma^{1/2}$, and the observed data matrix is taken as $X = \left(\Sigma^{1/2} Z^\top\right)^\top$. We then extract the largest two eigenvalues $\lambda_1 > \lambda_2$ of $\Sigma$ and the associated top eigenvector $v_1$ for evaluation. Figure A.1 shows the results for this dataset, which shows similar trends as the experiments described in Figure 2.

### F.2 Real data experiments

This section presents experiments on two real-world datasets. For each dataset, we show $\sin^2$ error with respect to the true offline eigenvector, used as a proxy for the ground truth, varying with the number of bits. The results are plotted in Figure A.2.

The goal of this section is to determine whether real-world experiments reflect the behavior of batched vs. standard methods with linear and logarithmic quantization. Therefore, we use the eigengap computed offline as a proxy of the true eigengap. If we wanted to compute the eigengap in an online manner, we could split the dataset randomly into a holdout set $\mathcal{S}$ and a training set $[n] \backslash \mathcal{S}$; run Oja's algorithm with quantization on a range of eigengaps with outputs $\mathbf{u}_1, \ldots, \mathbf{u}_m$, and select the one with the largest $\arg\max_i \mathbf{u}_i^T (\sum_{j \in S} \mathbf{D}_j \mathbf{D}_j^T) \mathbf{u}_i$ for a held out set $\mathcal{S}$.

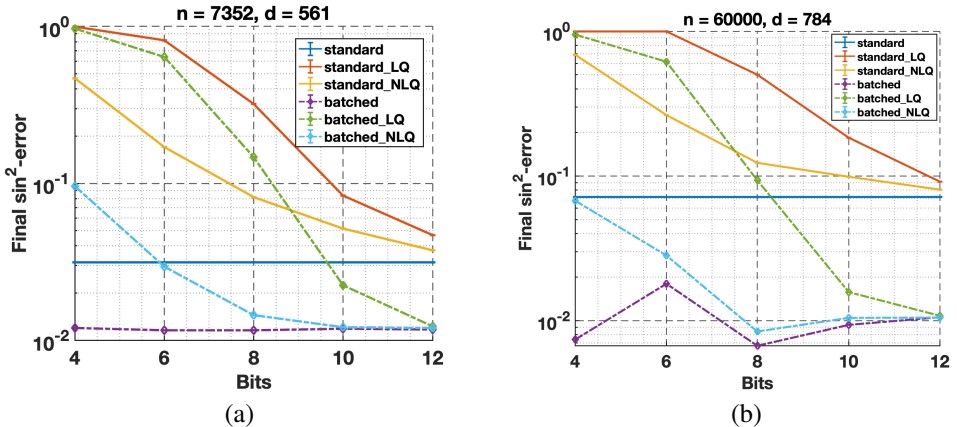

Figure A.2: Variation of $\sin^2$-error with bits for (a) HAR dataset (b) MNIST dataset.

**Time series + missing data**: The Human Activity Recognition (HAR) Dataset [AGO+13] contains smartphone sensor readings from 30 subjects performing daily activities (walking, sitting, standing, etc.). Each data instance is a 2.56-second window of inertial sensor signals represented as a feature vector. Here, $n = 7352$ and $d = 561$. For each datum, we also replace 10% of features randomly by zero to simulate missing data.

**Image data**: We use the MNIST dataset [LBBH98] of images of handwritten digits (0 through 9). Here, $n = 60,000, d = 784$, with each image normalized to a $28 \times 28$ pixel resolution.

These results collectively highlight that using the true offline eigengap (i) under stochastic rounding, batching provides a significant boost in performance since the quantization error does not depend linearly on $n$, and (ii) the logarithmic quantization attains a nearly dimension-free quantization error in comparison to linear quantization across a wide range of number of bits.

## G  Related Work

In this section, we provide some more related work on low-precision optimization. [DPHZ23] introduced QLoRA, which back-propagates through a frozen 4-bit quantized LLM into LoRA modules, enabling efficient finetuning of 65B-parameter models on a single 48 GB GPU with full 16-bit performance retention. Earlier works [XMHK23] examined the impact of stochastic round-off errors and their bias on gradient descent convergence under low-precision arithmetic. [YGG+24] propose *Collage*, a lightweight low-precision scheme for LLM training in distributed settings, combining block-wise quantization with feedback error to stabilize large-scale pretraining. Finally, communication-efficient distributed SGD techniques, such as 1-bit SGD with error feedback

[SFD$^+$14] and randomized sketching primitives (e.g., Johnson–Lindenstrauss projections [JL84]), further underscore the broad efficacy of low-precision computation.

**Low-Precision Optimization**: Reducing the bit-width of model parameters and gradient updates has proven effective for alleviating communication and memory bottlenecks in large-scale learning. QSGD [AGL$^+$17] uses randomized rounding to compress each coordinate to a few bits while preserving unbiasedness, incurring only an $O(\sqrt{d}/2^{\beta})$ increase in gradient noise for $\beta$ bits. [WXY$^+$17] maps gradients to $\{-1, 0, +1\}$ plus a shared scale and demonstrates negligible accuracy loss on ImageNet and CIFAR benchmarks. [SYKM17] achieve optimal communication–accuracy trade-offs via randomized rotations and scalar quantization. More recently, "dimension-free" analyses such as [LDS19] avoid scaling the required error rate with model dimension, instead depending on a suitably defined smoothness parameter.

