# OpenReview forum: "Low Precision Streaming PCA"
_NeurIPS.cc/2025/Conference — NeurIPS 2025 poster_

### Official Review · Reviewer_nH8S · 2025-06-24

**Clarity:** 2
**Significance:** 3
**Originality:** 3
**Rating:** 4
**Confidence:** 3

**Summary:**

Partially motivated by the advances of using limited precision arithmetic in neural networks, the authors study the effect of quantization on streaming PCA, restricted to finding the eigenvector with the largest eigenvalue. In particular, they analyze Oja's algorithm with two quantization schemes (linear and nonlinear), giving bounds on the "difference" of their approximated eigenvector and the optimal one. Additionally, the authors support their theoretical findings with a suite of experiments.

**Questions:**

Key points:

- Is the length of the stream $n$ fixed beforehand or can $n$ grow over time?
- Why there is no $\beta$ dependence in Lemma 1 and Lemma 2?
- What is meant by $\prod_{i=b}^1$ applied to a vector? E.g., line 143 and (7).
- In line 164, isn't everything on the l.h.s. constant and the assumption always holds? Or is $n$ increasing?
- Theorem 1/2: What is $\alpha$?

Further questions/remarks:

1. Do the authors require a bounded input set?
2. In line 59, is there no $n$ dependence for constructing and diagonalizing $\hat \Sigma$?
3. Can the authors clarify in the beginning, what they mean by "nearly dimension-free"? From my understanding, the dimension enters the error only logarithmically; this should be made explicit.
4. The norms in Assumption 1 are not made explicit: Is the expectation over the operator norm and the a.s. bound over the Frobenius norm?
5. In line 99, is $\beta \in \mathbb N_{>0}$?
6. In (3), what is the -1+1 for?
7. In the logarithmic scheme, line 107: Is it the same $\beta$ introduced for linear quantization?
8. In line 112, the authors state that they elaborate the stochastic rounding for linear quantization, but the equations concern $\mathcal Q_{NL}$, i.e., nonlinear quantization?
9. In line 113, can the authors make "well within the range" explicit?
10. In line 114, do $l$ and $u$ have index $i$? Or what do the $i$-s in (5) refer to?
11. In line 193, how explicitly were $\beta_e$ and $\beta_m$ chosen?
12. From which dataset was Figure 1 created? How was $b=10$ chosen? According to Theorem 2, $b$ depends on $n,\alpha$ and the eigengap $\lambda_1-\lambda_2$?
13. Algorithm 2: Line 5 does not depend on $i$, can it be moved out of the loop?

A few notes on typos/grammar:
- In line 46, is $\eta>0$ missing?
- In line 68, space before the reference.
- It could help to refer to the existence of the supplement in the outline (line 82-85).
- In line 87, it seems that $0\not \in [n]$; this should be made explicit, e.g., by using $\mathbb N_{>0}$ as $\mathbb N$ is ambiguous.
- Euclidean: Capitalize.
- In line 122, use "the norm of the estimated $\mathbf u$".
- Sometimes the authors use "Eq.", sometimes "equation", e.g., line 143, 151, 165.
- Line 156, $\mathbb R^{d\times d}$.
- Can the authors write "i.i.d." consistently?
- In line 167, "are bounded" => "is bounded".
- In line 168,  "in/see" duplicate.
- In line 205, add comma to $\min$.
- In line 220, "larger" => "smaller"?
- The bibliography is very inconsistent: Sometimes a venue has a number in words, other times in digits. Sometimes abbreviations are added, sometimes not. Some references have editors listed, others not. Names such as "Bernstein" should be capitalized. Similarly, abbreviations like "LLM", "DCM", "PCA".

**Ethical Concerns:**

["NO or VERY MINOR ethics concerns only"]

**Final Justification:**

Through consideration of the other reviewer's comments and the comprehensive rebuttal regarding my own review, I am bumping my score.

**Limitations:**

Yes.

**Paper Formatting Concerns:**

None.

**Quality:**

2

**Strengths And Weaknesses:**

### Strengths

- While PCA has been studied extensively in the literature, incorporating quantization and studying its effect is an original and novel twist.
- The lower bounds can be of independent interest, e.g., to compare future algorithms against.
- While the article is generally well written, I think that the presentation can be improved. See below.
### Weaknesses

- Some notations are not well-defined, rendering the equations challenging/impossible to parse/comprehend. See the questions for details.
- Similary, some abbreviations, e.g., BF16 in line 33 are not defined before they are used.
- A few existing related works are enumerated (Line 35--) but their actual results are not put into perspective.
- The practical relevancy, most prominently the impact on runtime, is not illustrated.
- The contributions are not clearly articulated. E.g., what does the "general theorem for streaming PCA" tell? Which lower bound (the new one or an existing one) is being matched?

---

> ### Author Rebuttal · Authors · 2025-07-30
>
> Thank you for your kind words regarding the originality, novelty, and the writing of the manuscript. We will fix all typographical errors pointed out by the reviewer in the revised manuscript and address key questions below:
>
> **Regarding bfloat16** : The bfloat16 (brain floating point) floating-point format uses 16 bits, designed for efficient deep learning training and inference. We will clarify this in the paper.
>
> **Regarding related work** : We provide more details on related work in Appendix G. However, most of the known guarantees are for stochastic gradient descent (SGD) with convex objectives, and we are not aware of existing analyses that apply to the streaming PCA setting off the shelf.
>
> **Regarding runtime comparisons** : When operating with $\beta$-bits, the overall complexity for streaming PCA (and that of the batched variant) is $O(nd\beta)$ and we performed a benchmarking experiment to illustrate this below.
>
> | Experiment | Runtime (seconds) |
> | ---------- | -------------------- |
> | 64 bits  | 0.0274 ± 0.00136  |
> | 32 bits  | 0.0260 ± 0.00225  |
> | 16 bits  | 0.000398 ± 0.0000235 |
>
> **Regarding contributions** : The general theorem refers to a result for streaming PCA with Oja’s algorithm that can handle martingale noise in the iterates, and is a generalization of known analysis with IID data. We are unaware of existing lower bounds for our problem setting and refer to the lower bounds proved in our work. We will clarify this.
>
> **Regarding length of the stream**: In the current manuscript, the length of the stream $n$ is an input and the learning rate is constant over time for ease of analysis. To handle variable learning rates using only constant-rate updates, we employ a standard \emph{doubling trick}. Specifically, we divide the time horizon into blocks of exponentially increasing size, where each block $k$ has length $2^k$ and uses a constant learning rate $\eta_k$ tailored to that block’s length. At the end of each block, the previous estimate is discarded and a new estimate is initialized. This scheme effectively simulates a decaying learning rate $\eta_t$ while retaining the simplicity and analytical tractability of constant-rate updates.
>
> **Regarding $\beta$ dependence in Lemma 1 and 2**: While the bounds in Lemma 1 and 2 are stated for a given $\delta, d$, the optimal choice of these parameters given a fixed bit budget $\beta$ does make the bounds dependent on \beta (see lines 133-137).
>
> **$\prod_{i=b}^{1}$ applied to a vector**: The product is a product of matrices in order $b$ to $1$, which is then multiplied by a vector. We will clarify this in the revised manuscript.
>
> **Regarding L.H.S of line 164**: The number of batches $b$ depends on $n$. The $O(1)$ factor here represents a constant w.r.t asymptotic behaviour of n. We will clarify this in the revised manuscript.
>
> **Regarding $\alpha$ in Theorem 1 and 2**: $\alpha$ is a constant which determines the learning rate, $\eta$. Our results require $\alpha \geq 1$ (see Lemma A.2). We will refer to this lemma in the main paper.
>
> **Regarding bounded input set assumption**: We only require the variance and bounded assumptions in Assumption 1, which is popular in streaming PCA literature (see [1-7]). We note that while the current analysis assumes a bounded set, this is for ease of analysis and it can be generalized to unbounded subgaussian data (see [8, 9, 10]).
>
> For space constraints and ease of exposition, we will provide an easy argument that changes the algorithm by considering "truncated" datapoints $Y_i = X_i \mathbb{1}(||X_{i}||^{2} \leq \alpha_n)$, where the truncation parameter $\alpha_n$  is set to be $c\log(n)$. Thus, we replace any datapoint whose squared norm exceeds the truncation parameter by zero. Standard truncation arguments can then show that the sin-squared error of the output of this algorithm w.r.t the original principal eigenvector has the same theoretical guarantee. The crux of the argument is that the covariance matrix of the truncated data changes very little since the truncations happen with very small probability.
>
> **Regarding time and space complexity for diagonalizing $\hat{\Sigma}$** : Thank you for pointing this out - our intention was to say that the time and space dependencies of offline diagonalization are $\Omega(d^2)$. We will clarify this.
>
> **Regarding “nearly dimension-free” bounds** : Yes, our dimension dependence is logarithmic in $d$, which we refer to as nearly dimension-free and we will make this explicit in the revised manuscript.
>
>  **Regarding $||.||$ norm** : Both norms are the operator norm $||.||_2$. Throughout the paper, the norms $||.||$, $||.||_2$ and $||.||\_{op}$ all refer to the 2-norm $||.||_2$ (which is the operator norm for matrices). We will clarify this in the notation section.
>
> **Regarding $\beta$ being natural number** : Yes, $\beta$ is the number of bits used by the model and is a natural number.
>
> **Regarding +1,-1 in (3)** : Yes, the second value in the quantization should be $-\delta (2^{\beta-1} - 1)$ instead of $-\delta ((2^{\beta-1}-1)+1).$ We will fix this typo.
>
> **Regarding stochastic rounding in Line 112**: Thank you for pointing that out. We will fix this typographical error.
>
> **Regarding “well within the range”**: Thank you for pointing it out. Here, we mean that the value being rounded is between the smallest and the largest numbers in the quantization grid.
>
> **Regarding $\ell$ and $u$ in Line 114**: Thank you for pointing out the typographical error. The i's in (5) will be removed in the revised paper.
>
> **Regarding choice of $\beta_e$ and $\beta_m$**: The choice of the parameters $\zeta$ and $\delta_0$ is inspired from floating point computation, where the choice of the number of bits in the mantissa, $\beta_m$, determines $\zeta$ and the number of bits in the exponent, $\beta_e$, determines the smallest representable number $\delta_0$. The values $\beta_e$ and $\beta_m$ were chosen by minimizing the error $\kappa_1 = \zeta^2 + \delta_0^2 d$ explicitly, that is, by taking a derivative with respect to $\beta_e$ and setting the expression to zero. Note that the quantization errors $\zeta$ and $\delta_0$ depend on the bits and the choice is inspired from floating point computation.
>
> **Regarding the dataset used in Figure 1**: The dataset is the same as used in Section 5, Experiments and is described in detail in Appendix F. Since the dataset is synthetically generated, we are able to explicitly calculate the eigengap. However, we would like to point out that the analysis is robust to upper bounds off by a multiplicative constant on the eigengap with a slight increase in the final sin-squared error.
>
> **Regarding Algorithm 2, line 5**: Yes, we will move the definition of $\rho$ outside the loop, making the algorithm more readable.
>
> ### **References**
> [1] Moritz Hardt and Eric Price. The noisy power method: a meta algorithm with applications. NeurIPS 2014.
>
> [2] Christopher De Sa, Christopher Re, and Kunle Olukotun. Global convergence of stochastic gradient descent for some non-convex matrix problems. ICML 2015.
>
> [3] Ohad Shamir. Convergence of stochastic gradient descent for PCA. ICML 2016.
>
> [4] Ohad Shamir. Fast stochastic algorithms for SVD and PCA: convergence properties and convexity. ICML 2016.
>
> [5] Zeyuan Allen-Zhu and Yuanzhi Li. First efficient convergence for streaming k-PCA: a global gap-free and near-optimal rate. FOCS, 2017.
>
> [6] Prateek Jain, Chi Jin, Sham M. Kakade, Praneeth Netrapalli, and Aaron Sidford. Streaming PCA: Matching matrix Bernstein and near-optimal finite sample guarantees for Oja’s algorithm. COLT, 2016.
>
> [7] Maria-Florina Balcan, Simon Shaolei Du, Yining Wang, and Adams Wei Yu. An improved gap-dependency analysis of the noisy power method. COLT, 2016
>
> [8] Lunde, R., Sarkar, P. and Ward, R., 2021. Bootstrapping the error of Oja's algorithm. Advances in neural information processing systems, 34, pp.6240-6252.
>
> [9] Liang, X., 2023. On the optimality of the Oja’s algorithm for online PCA. Statistics and Computing, 33(3), p.62.
>
> [10] Kumar, S. and Sarkar, P., 2024. Oja's algorithm for streaming sparse PCA. Advances in Neural Information Processing Systems, 37, pp.74528-74578.

---

> > ### Comment · Reviewer_nH8S · 2025-08-01
> >
> > I thank the authors for considering and answering my questions to my full satisfaction. In particular, with the $\Pi_{i=b}^1$ issue (which apparently also confused another reviewer) settled and the choice of $\beta$-s clarified, I am happy to bump my score and trust that the authors incorporate the content of their rebuttals into the revised version of the manuscript.

---

> > > ### Author Response · Authors · 2025-08-01
> > >
> > > Thank you for your positive response. We will incorporate these points and clarify the choice of $\beta$ in the revised manuscript.

---

### Official Review · Reviewer_6QDR · 2025-06-28

**Clarity:** 3
**Significance:** 3
**Originality:** 3
**Rating:** 5
**Confidence:** 3

**Summary:**

The authors proposed a low-precision version Oja's algorithm for streaming PCA, in which the iterates and the samples and the gradients are all quantized, and showed that its batched version achieves optimal estimation error rates, under either linear quantizer (LQ) and nonlinear quantizer (NLQ). The take-away lies on the advantage of NLQ in achieving nearly dimension-free estimation error.

**Questions:**

1. Can the authors further clarity the motivations/applications of their problem setting?  I am a bit confused about the "quantization constraint" -- specifically, what kind of quantization information one can use to estimate $v_1$ in the authors' setting.

In the algorithm, the authors quantize everything (samples, gradient, iterates) -- is this motivated by some practical regimes, or can we only quantize the samples?  Even only quantizing the samples, the problem distinguishes with some problems I know of where the samples should be quantized before running the algorithms (e.g., see "Distributed gaussian mean estimation under communication constraints: Optimal rates and communication-efficient algorithms").

2. The phenomenon that NLQ is statistically much better than LQ is new to me. I understand that this stems from the choice of the relevant parameters -- which nonetheless appears less straightforward to the readers.

Can the authors provide some intuition that leads to such difference? Are there similar results appearing in other low-precision estimation problems?

3. The authors studied two specific quantizers. Can the authors' method work other quantizers? Or can the authors explain why they specifically consider these two quantizers?

**Ethical Concerns:**

["NO or VERY MINOR ethics concerns only"]

**Final Justification:**

I maintain my current rating. The authors' responses address my main concerns and promote my understanding to their work. The setting differs from a standard estimation problem under quantization, in that everything is quantized throughout the computational procedure. The results are novel and interesting.

**Limitations:**

The limitation has been listed as questions above.

**Quality:**

3

**Strengths And Weaknesses:**

Strengths:

1) The paper conveys rather clear messages - NLQ achieves faster nearly dimension-free rate - and batched version (or sample splitting) leads to much better performance.

2) The results provide a clear understanding on the problem - the authors established lower bounds and matching upper bounds (by quantized batched Oja's). In view of Sec. 4, the proof techniques are nontrivial with some novelty.

3) The experiments nicely back up the theory.

Weaknesses:

1) Some ambiguity in notation: P3 only defines $\|.\|_2,\|.\|_{op}$, while later $\|.\|$ is frequently used (e.g., Assumption 1, Eqs. 8-9)
2) The weakness is that the motivation of such study should be made clearer.

---

> ### Author Rebuttal · Authors · 2025-07-30
>
> Thank you for your valuable feedback on the clarity of our paper and the novelty of our results. We will clarify the motivation of our work and make our notations consistent in a revision.
>
> ## Motivation
> Hardware accelerators—TPUs, for instance—derive much of their speed advantage from relying on low-precision arithmetic. In this paper, we analyze streaming PCA in a low-precision setting, where intermediate values stored in memory have a limited number of bits. This leads to quantization errors at every intermediate step [1-4].  We model this behavior as follows: (i) quantize at points where any computed values are stored in memory, and (ii) fundamental arithmetic operations such as addition, multiplication, dot products, and matrix-vector products are quantized after being computed in high precision.
> Note that this is different from only quantizing the samples, as the latter allows computed quantities to be stored in high precision.
> When operating with $\beta$-bits, the overall complexity for streaming PCA (and that of the batched variant) is $O(nd\beta)$. We performed a benchmarking experiment to illustrate this below.
>
> | Experiment | Runtime (seconds) |
> | ---------- | -------------------- |
> | 64 bits  | 0.0274 ± 0.00136  |
> | 32 bits  | 0.0260 ± 0.00225  |
> | 16 bits  | 0.000398 ± 0.0000235 |
>
> ## NLQ is statistically much better than LQ
> Non-linear quantization (NLQ) is a type of logarithmic scheme where adjacent points in the quantization grid are multiplicatively close, that is, the error due to quantization in NLQ is proportional to the value itself. On the other hand, the linear quantization (LQ) scheme has an additive error, which can be (i) lossy when the value being quantized is small (see [2]), and (ii) needlessly accurate if the values are large.
>
> In Oja’s algorithm with quantization, the learning rate is small, and the updates to the values are themselves small. If the error due to quantization is additive and of the same order as the value itself, then the direction of the gradient is significantly different compared to when the coordinates of the gradient directions are correct up to multiplicative factors.
>
> ## Regarding Quantization Schemes
> The quantization schemes we use have been widely used by other works [2-5]. As we discuss below, our analysis can be readily adapted to the following quantization schemes commonly used in low-precision computing.
>
> The Floating Point Quantization (FPQ) is widely adopted and is a type of Logarithmic quantization scheme, where adjacent values in the quantization grid are multiplicatively close. Logarithmic schemes are of wide practical interest and are used in most modern programming languages such as C++, Python, and MATLAB, and are standardized for common use (such as the IEEE 754 floating point standard [6]). Another quantization scheme used for low-precision training is power-of-two quantization [1], where any value is rounded down to the nearest power of two. This is yet another type of logarithmic quantization.
> Logarithmic quantization is similar in principle to the non-linear quantization (NLQ) scheme we use in our work. In particular, Lemma A.9 in the appendix establishes a relationship between the distance of a quantized vector and the original one under NLQ. This Lemma applies to FPQ and more generally to most other logarithmic quantization schemes. We believe that **our proofs can be modified to work with any logarithmic scheme, including FPQ**.
>
> ### **References**
>
> [1] Przewlocka-Rus, Dominika, et al. "Power-of-two quantization for low bitwidth and hardware compliant neural networks." arXiv preprint arXiv:2203.05025 (2022).
>
> [2] Courbariaux, Matthieu, Yoshua Bengio, and Jean-Pierre David. "Training deep neural networks with low precision multiplications, 2014."
>
> [3] Li, Zheng, and Christopher M. De Sa. "Dimension-free bounds for low-precision training." NeurIPS 2019.
>
> [4] De Sa, Christopher, et al. "High-accuracy low-precision training."
>
> [5] Das, Dipankar, et al. "Mixed precision training of convolutional neural networks using integer operations." ICLR 2018.
>
> [6] Kahan, William. "IEEE standard 754 for binary floating-point arithmetic."

---

> > ### Comment · Reviewer_6QDR · 2025-08-06
> >
> > Thanks for the responses which address my main concern in a satisfactory manner. I agree with the authors that the motivation of their study is different from those quantizing the samples only before the statistical procedure. I will maintain my score.

---

### Official Review · Reviewer_3qrH · 2025-07-02

**Clarity:** 3
**Significance:** 4
**Originality:** 4
**Rating:** 4
**Confidence:** 4

**Summary:**

This paper presents a rigorous study of low-precision algorithms for streaming principal component analysis (PCA). The motivation stems from the increasing need to perform high-dimensional data processing under severe memory and bandwidth constraints. The authors propose variants of Oja’s algorithms that apply quantization to either gradients or data, and they analyze their convergence properties under both linear and logarithmic stochastic quantizations.

1. Quantized Oja's Algorithm: The authors generalize Oja's method to the setting where the data or gradients are low-precision representations, and rigorously analyze its convergence to the leading eigenvector of the data covariance matrix.
3. Theoretical Guarantees: They provide non-asymptotic convergence bounds under various quantization models (e.g., unbiased stochastic quantizers and one-bit compression), and establish trade-offs between quantization level, convergence rate, and variance.
4. Lower Bound: A matching lower bound is provided for the streaming PCA problem under quantization constraints, characterizing the inherent difficulty of the task.
5. Empirical Validation: Experiments on synthetic and real-world datasets validate the theoretical findings, demonstrating that low-precision algorithms can achieve nearly the same accuracy as full-precision ones at a fraction of the communication cost .

**Questions:**

--
Why do you need to quantize every step of the algorithm? Why can’t you compute a final update and quantize that? In particular, I find what is the purpose of the quantization in Line~5, Algorithm~1?

**Ethical Concerns:**

["NO or VERY MINOR ethics concerns only"]

**Limitations:**

--
Use of excessive hyperparameters whose knowledge might not be known priori.

**Quality:**

3

**Strengths And Weaknesses:**

Strengths:
This is a good algorithm for streaming PCA with low computational precision. The particular novelty lies in designing the algorithm (as in when to quantize). The probability boosting framework is standard and of course, it comes at the cost of additional space and memory.
The proofs follow a standard decomposition of the projection error and matrix product concentration.

Weakness:

--
Some theoretical results depend heavily on simplifying assumptions about data distributions or quantization noise, and may not fully reflect practical scenarios (e.g., non-iid or adversarial data). See for example, Robust Streaming PCA NeurIPS 2022, which considers one such practical model for this problem.

--
The learning rate requires knowledge of spectral gap. This is unreasonable in practice.

---

> ### Author Rebuttal · Authors · 2025-07-30
>
> Thank you for your helpful feedback and for recognizing the novelty in our work.
>
> ## Assumptions about data distribution, quantization noise, and non-IID/adversarial data
> Below we sketch that the assumptions we use are widely adopted in streaming PCA and quantization literature. We also provide a discussion of how to adapt our techniques to non-IID settings.
>
> ### **Assumptions about data distributions**
>
> Assumption 1 states standard moment bounds used to analyze PCA in the stochastic setting. These assumptions are also used in citations [5-12] to derive near-optimal sample complexity bounds for Oja’s rule.
>
> ### **Quantization noise**
>
> The quantization schemes we use have been widely used by other works [1,2,3,4]. As we discuss below, our analysis can be readily adapted to the following quantization schemes commonly used in low-precision computing.
>
> The Floating Point Quantization (FPQ) is widely adopted and is a type of Logarithmic quantization scheme, where adjacent values in the quantization grid are multiplicatively close. Logarithmic schemes are of wide practical interest and are used in most modern programming languages such as C++, Python, and MATLAB, and are standardized for common use (such as the IEEE 754 floating point standard [19]). Another quantization scheme used for low-precision training is power-of-two quantization [16], where any value x is rounded down to the nearest power of two. This is yet another type of logarithmic quantization.
> Logarithmic quantization is similar in principle to the non-linear quantization (NLQ) scheme we use in our work. In particular, Lemma A.9 in the appendix establishes a relationship between the distance of a quantized vector and the original one under NLQ. This Lemma applies to FPQ and more generally, to most other logarithmic quantization schemes. We believe that **our proofs can be modified to work with any logarithmic scheme**.
>
> ### **Non-IID or adversarial data**
>
> In the streaming PCA literature, we are aware of the following extensions to non-IID data.
>
> **Independent but non-identically distributed data**: Robust Streaming PCA ([3]) handles data generated from covariance matrices that change over time (while being in the family of spiked models with $k$ spikes). Their analysis tracks the error accumulated from the mismatch of the varying covariance matrices with the last one in each batch. We believe that our rounding algorithm and analysis can be used to replace the part that considers matrix concentration.
>
> **Markovian datasets**: [18] analyzes Oja’s algorithm for Markovian datasets. We believe that our analysis, which incorporates martingale noise resulting from quantization, can be easily combined with their analysis to obtain results for Markovian data.
>
> **Adversarial or contaminated data**: The proof techniques here are vastly different ([13,14,17]) and would require substantially different tools. This presents an interesting direction for future work.
>
> We believe that our work represents a **first step in exploring quantized streaming PCA** algorithms, which can be built upon to analyze various extensions.
>
> ## Hyperparameters and learning rate
>
> **Only two hyperparameters**: Parameters such as $\mathcal{V}$ and $\mathcal{M}$, and quantization parameters $\delta, \alpha, \delta_0$, are required for analysis only. **Hence, these are not hyperparameters.**
>
> The only two hyperparameters required by Algorithm 1 are the batch size $b$ and the learning rate $\eta$. The optimal choice for both is determined by the spectral gap $\gamma = \lambda_1 - \lambda_2$.
>
> **Learning rate and eigengap**: The expression for the learning rate $\eta = \frac{\alpha \log n}{n (\lambda_1 - \lambda_2)}$ is also present in other works on streaming PCA [5-12] to derive the statistically optimal sample complexity (up to logarithmic factors). In particular, Theorem 1 (lines 165-166) in our paper shows that the error in the output vector is
>
> $d e^{-n\eta (\lambda_1-\lambda_2)}+\frac{\eta \mathcal{V}}{\lambda_1-\lambda_2}+\max(\frac{b}{\alpha\log n}, 1) \kappa_1. $
>
> If a smaller learning rate $\eta$ is used (for example, from plugging in an upper bound $U$ on the eigengap $(\lambda_1 - \lambda_2)$ and using a crude upper bound $\eta = \frac{\alpha \log n}{n U}$ instead of $\eta = \frac{\alpha \log n}{n (\lambda_1 -\lambda_2)}$ will make the first term larger, leading to a slightly larger sin-squared error. A similar argument shows that the batch size is also robust to the choice of an upper bound on the eigengap. We will clarify this further in the revised manuscript.
>
> ## Quantization at every step of the algorithm
>
> Our motivation for this study was to study the theoretical convergence guarantess of PCA in low-precision settings, which improves runtimes in practice. In the low-precision setting, intermediate values stored in memory have a limited number of bits, leading to quantization errors at every intermediate step [1-4]. We model this behavior by (i) quantizing at points where computed values are stored in memory, and (ii) fundamental arithmetic operations such as addition, multiplication, dot products, and matrix-vector products are quantized after being computed in high precision. In Line 5, Algorithm 1, the intermediate summands are quantized because they are stored in memory. We quantize again after summing the values.
>
> While in some settings, it may be reasonable to quantize at the final step, in this paper, we analyze streaming PCA in a low-precision setting, where intermediate values stored in memory have a limited number of bits. When the quantization is applied at the last step, the analysis is straightforward since the final error is equal to the sum of the error without quantization and the additional error $\delta$ introduced by quantizing the final result.
>
> ## References
>
> [1] Courbariaux, Matthieu, Yoshua Bengio, and Jean-Pierre David. "Training deep neural networks with low precision multiplications, 2014."
>
> [2] Li, Zheng, and Christopher M. De Sa. "Dimension-free bounds for low-precision training." NeurIPS 2019.
>
> [3] De Sa, Christopher, et al. "High-accuracy low-precision training."
>
> [4] Das, Dipankar, et al. "Mixed precision training of convolutional neural networks using integer operations." ICLR 2018.
>
> [5] Moritz Hardt and Eric Price. The noisy power method: a meta algorithm with applications. NeurIPS 2014.
>
> [6] Christopher De Sa, Christopher Re, and Kunle Olukotun. Global convergence of stochastic gradient descent for some non-convex matrix problems. ICML 2015.
>
> [7] Ohad Shamir. Convergence of stochastic gradient descent for PCA. ICML 2016.
>
> [8] Ohad Shamir. Fast stochastic algorithms for SVD and PCA: convergence properties and convexity. ICML 2016.
>
> [9] Zeyuan Allen-Zhu and Yuanzhi Li. First efficient convergence for streaming k-PCA: a global gap-free and near-optimal rate. FOCS, 2017.
>
> [10] De Huang, Jonathan Niles-Weed, and Rachel Ward. Streaming k-PCA: Efficient guarantees for Oja’s algorithm, beyond rank-one updates. COLT, 2021.
>
> [11] Prateek Jain, Chi Jin, Sham M. Kakade, Praneeth Netrapalli, and Aaron Sidford. Streaming PCA: Matching matrix Bernstein and near-optimal finite sample guarantees for Oja’s algorithm. COLT, 2016.
>
> [12] Maria-Florina Balcan, Simon Shaolei Du, Yining Wang, and Adams Wei Yu. An improved gap-dependency analysis of the noisy power method. COLT, 2016
>
> [13] Diakonikolas, Ilias, et al. "Nearly-linear time and streaming algorithms for outlier-robust PCA." ICML 2023.
>
> [14] Cheng, Yu, Ilias Diakonikolas, and Rong Ge. "High-dimensional robust mean estimation in nearly-linear time." SODA 2019.
>
> [15] Bienstock, Daniel, et al. "Robust streaming PCA." NeurIPS 2022.
>
> [16] Przewlocka-Rus, D., Sarwar, S.S., Sumbul, H.E., Li, Y. and De Salvo, B., 2022. Power-of-two quantization for low bitwidth and hardware compliant neural networks. arXiv preprint arXiv:2203.05025.
>
> [17] Price, E. and Xun, Z., 2024, October. Spectral guarantees for adversarial streaming PCA. FOCS 2024.
>
> [18] Kumar, S. and Sarkar, P.. Streaming PCA for Markovian data. NeurIPS 2023.
>
> [19] Kahan, William. "IEEE standard 754 for binary floating-point arithmetic."

---

> > ### Comment · Reviewer_3qrH · 2025-08-05
> >
> > Thank you for your reply. I will maintain my score.

---

### Official Review · Reviewer_Bmfn · 2025-07-02

**Clarity:** 2
**Significance:** 3
**Originality:** 2
**Rating:** 4
**Confidence:** 3

**Summary:**

The paper gives quantized version of Oja's algorithm for streaming PCA under both linear and logarithmic quantization schemes along with randomized rounding. They give lower bounds on quantization error for both quantization schemes. The batched versions of their algorithms almost match the lower bounds. The Non logarithmic quantization scheme can achieve errors which are independent of the dimension and hence very useful in high dimensions. The authors also give a way to boost the success probability of their algorithm. They validate their guarantees by experiments on synthetic data.

**Questions:**

See Weaknesses

**Ethical Concerns:**

["NO or VERY MINOR ethics concerns only"]

**Final Justification:**

I have modified my score base on rebuttal. But I would urge the authors to improve the presentation and explicitly explain the multiplications happening to remove confusions

**Limitations:**

Yes

**Paper Formatting Concerns:**

See weaknesses

**Quality:**

2

**Strengths And Weaknesses:**

Strengths:
1) PCA is a very important problem in machine learning. Addressing the problem of finding out principal components in low precision scenario is an important and interesting problem.
2)Lower bounds for quantization errors for linear and nonlinear quantization schemes and algorithms that almost matches the lower bounds.
3) The algorithm is easy to understand and appears simple to implement too.
4) I did not check the proofs in detail however the results appear correct and sound.

Weaknesses and Suggestions:
1) Overall, I did not like the writing of the paper. Better notational convention can be used. Also, there are typos. For e.g. I think there is a misplaced or additional bracket in eq. (3). Please correct me if I am wrong. In all equations involving product over terms, e.g. eq. (7) why is the lower limit of product $b$ and upper limit $1$. Again, it appears weird to me unless I missed something.
2) Experiments are only performed on small set of synthetic data.

---

> ### Author Rebuttal · Authors · 2025-07-30
>
> Thank you for your valuable feedback. We will fix all typographical errors in the revision; in particular, the second value in the set of equation (3) should be $-\delta \cdot (2^{\beta-1}-1)$ instead of $-\delta \cdot ((2^{\beta-1}-1)+1)$.
>
> ## Lower and upper limits in matrix product
>
> Each iteration of the Oja’s algorithm has a gradient update $u_{t+1}=(I+\eta X_{t+1}X_{t+1}^T)u_{t}$ followed by normalization $u_{t+1} = u_{t+1}/\left\lVert u_{t+1} \right\rVert$. As a result, the final vector $u_T$ be unwound into the (normalized) product of n matrices successively left-multiplied to the initial vector $u_0$.
> The crucial observation here is that the first term in the matrix from the left uses the _last_ data point, while the rightmost matrix corresponds to the _first_ data point. This is why the lower and upper limits go backward. We will clarify this notation in the revision.
>
> ## Experiments
>
> We conducted additional experiments on both synthetic and real-world datasets. For real-world datasets, we consider the top eigenvector of the sample covariance matrix of the data as the ground truth. All experiments below show the $\sin^2$ error with respect to the ground truth eigenvector. For all three datasets, **we observe that the trends of these experiments are similar to those presented in our paper**. In particular, we observe that, in general,
>
> 1) All quantized methods perform better as the bit budget increases.
> 2) Batched methods outperform their analogues with per-iterate updates.
> 3) NLQ outperforms LQ for a given bit budget.
>
> ### **Additional Synthetic Data**
>
> Our first experiment considers covariance matrices generated using a random orthonormal basis $Q$, chosen by performing QR decomposition of $Z$ such that $Z_{i,j} \sim N(0,1)$ with eigenvalues decaying as $\lambda_{i} := i^{-2}$. We use $n=10,000$ and $d=100$.
>
> | Method             | 4 bits           | 6 bits           | 8 bits           | 10 bits          | 12 bits          |
> | ------------------ | ---------------- | ---------------- | ---------------- | ---------------- | ---------------- |
> | Standard Oja       | 0.004 ± 5.92e-17 | 0.004 ± 3.63e-17 | 0.004 ± 2.22e-17 | 0.004 ± 5.97e-17 | 0.004 ± 2.96e-17 |
> | Standard Oja + LQ  | 0.960 ± 0.017    | 0.454 ± 0.019    | 0.109 ± 0.005    | 0.016 ± 7.16e-04 | 0.005 ± 1.55e-04 |
> | Standard Oja + NLQ | 0.297 ± 0.010    | 0.117 ± 0.005    | 0.030 ± 0.001    | 0.010 ± 4.98e-04 | 0.006 ± 1.36e-04 |
> | Batched Oja        | 0.006 ± 0.004    | 0.002 ± 0.002    | 0.001 ± 6.13e-04 | 0 ± 1.99e-06     | 0 ± 7.21e-08     |
> | Batched Oja + LQ   | 0.572 ± 0.013    | 0.068 ± 0.002    | 0.005 ± 1.55e-04 | 0 ± 2.03e-05     | 0 ± 3.14e-06     |
> | Batched Oja + NLQ  | 0.033 ± 0.001    | 0.005 ± 2.41e-04 | 0.001 ± 3.47e-05 | 0 ± 1.58e-05     | 0 ± 5.01e-06     |
>
> ### **Image data**
>
> We use the **MNIST dataset**, which comprises grayscale images of handwritten digits (0 through 9). Here, n = 60,000, d = 784, with each image normalized to a 28 × 28 pixel resolution.
>
> | Method             | 4 bits           | 6 bits           | 8 bits           | 10 bits          | 12 bits          |
> | ------------------ | ---------------- | ---------------- | ---------------- | ---------------- | ---------------- |
> | Standard Oja       | 0.063 ± 1.22e-10 | 0.063 ± 2.13e-10 | 0.063 ± 1.94e-10 | 0.063 ± 1.90e-10 | 0.063 ± 2.66e-10 |
> | Standard Oja + LQ  | 0.997 ± 0.002    | 0.93 ± 0.009     | 0.421 ± 0.008    | 0.143 ± 0.002    | 0.07 ± 0.001     |
> | Standard Oja + NLQ | 0.511 ± 0.030    | 0.226 ± 0.008    | 0.110 ± 0.003    | 0.072 ± 9.98e-04 | 0.066 ± 5.31e-04 |
> | Batched Oja        | 0.372 ± 0.106    | 0.268 ± 0.081    | 0.270 ± 0.095    | 0.021 ± 0.026    | 0.020 ± 0.012    |
> | Batched Oja + LQ   | 0.887 ± 0.010    | 0.273 ± 0.007    | 0.050 ± 8.32e-04 | 0.030 ± 2.76e-04 | 0.030 ± 2.15e-04 |
> | Batched Oja + NLQ  | 0.067 ± 0.056    | 0.034 ± 5.57e-04 | 0.030 ± 2.62e-04 | 0.029 ± 1.49e-04 | 0.031 ± 2.11e-04 |
>
> ### **Time series data**
>
> The **Human Activity Recognition (HAR) dataset** contains smartphone sensor readings from 30 subjects performing daily activities (walking, sitting, standing, etc.). Each data instance is a 2.56-second window of inertial sensor signals represented as a feature vector. We have (n = 7,352, d = 561), and the data constitutes a time series.
>
> | Method             | 4 bits           | 6 bits           | 8 bits           | 10 bits          | 12 bits          |
> | ------------------ | ---------------- | ---------------- | ---------------- | ---------------- | ---------------- |
> | Standard Oja       | 0.013 ± 3.19e-10 | 0.013 ± 1.56e-09 | 0.013 ± 4.12e-10 | 0.013 ± 4.31e-10 | 0.013 ± 3.74e-10 |
> | Standard Oja + LQ  | 0.998 ± 2.33e-04 | 0.871 ± 0.016    | 0.305 ± 0.003    | 0.087 ± 0.002    | 0.028 ± 5.94e-04 |
> | Standard Oja + NLQ | 0.455 ± 0.014    | 0.157 ± 0.004    | 0.068 ± 0.001    | 0.033 ± 9.53e-04 | 0.020 ± 3.96e-04 |
> | Batched Oja        | 0.822 ± 0.025    | 0.670 ± 0.075    | 0.053 ± 0.053    | 0.063 ± 0.005    | 0.005 ± 0.010    |
> | Batched Oja + LQ   | 0.943 ± 0.005    | 0.405 ± 0.006    | 0.041 ± 0.001    | 0.009 ± 1.05e-04 | 0.006 ± 3.60e-05 |
> | Batched Oja + NLQ  | 0.047 ± 5.58e-04 | 0.013 ± 1.86e-04 | 0.007 ± 8.97e-05 | 0.007 ± 2.64e-05 | 0.006 ± 1.33e-05 |

---

> > ### Comment · Reviewer_Bmfn · 2025-08-05
> > **Reply**
> >
> > Thank you for the rebuttal. I have modified my score

---

### Official Review · Reviewer_4RCk · 2025-07-03

**Clarity:** 3
**Significance:** 4
**Originality:** 4
**Rating:** 5
**Confidence:** 3

**Summary:**

The authors establish new lower bounds on linear and non-linear logarithmic quantized Oja's algorithm for streaming PCA. The bound is sharp (up to logarithmic factors) when batching is used. Error for log quantization is dimension-free, while linear quantization scaled with the dimension of the data points. The authors conduct empirical analysis over varying sample size, data dimension, and quantization bits, along with guidance for selecting optimal quantization hyperparameters when given a bit budget.

**Questions:**

1. What are the core challenges for extension to top-k? Namely, what prevents analysis over the deflated covariance matrix (once the previous component is found)?

2. Is assumption 1 unreasonable? I'm not asking because I believe it's unreasonable, I just do not know if it's used in other similar results besides [1]. For example, streaming convergence for the power method with momentum in [2] does not seem to rely on assumptions regarding the distributions of entries in the X_i.

[1] Kumar, S., & Sarkar, P. (2024). Oja's Algorithm for Streaming Sparse PCA. arXiv preprint arXiv:2402.07240.

[2] Xu, P., He, B., De Sa, C., Mitliagkas, I., & Re, C. (2018, March). Accelerated stochastic power iteration. In International Conference on Artificial Intelligence and Statistics (pp. 58-67). PMLR.

**Ethical Concerns:**

["NO or VERY MINOR ethics concerns only"]

**Final Justification:**

I am maintaining my score. I find this bound interesting as this result seemingly doesn't already exist in literature for such a classical method (Oja's method). At least one reviewer has complained that some knowledge of the spectral gap is required but observe that is the case with similar literature [1,2]. Spectral-free knowledge which results in accelerated convergence seems notoriously difficult to acquire.

[1] Xu, Peng, et al. "Accelerated stochastic power iteration." International Conference on Artificial Intelligence and Statistics. PMLR, 2018.

[2] Rabbani, Tahseen, et al. "Practical and fast momentum-based power methods." Mathematical and Scientific Machine Learning. PMLR, 2022.

**Limitations:**

yes

**Quality:**

4

**Strengths And Weaknesses:**

Strengths:

 - A novel lower bound for a very classical algorithm (Oja + streaming PCA) with strong applications to machine learning. Indeed, it surprises me that such a result for quantized streaming PCA using Oja's rule had not already been established in literature. Bounds for both linear and logarithmic are provided, which enables flexibility on quantization procedure.

 - Very easy to read -- the review of streaming PCA, Oja's algorithm, and quantization + stochastic rounding will be comprehensible for those outside of numerical optimization.

 - The lower bound is sharp (up to algorithmic factors) when batching is used, demonstrating the utility of the bound.

 - Explicit determination of optimal quantization parameters under a fixed budget makes these bounds practically useful as well.

 - Reduction of linear dependence on dataset size to logarithmic dependence via batching is a nice result.

Weaknesses

 - Results do not apply to extraction of top-k components.

 - I find the proof techniques in the manuscript do not necessarily reveal enough intuition for the core results. Deeper proof sketches of the core theorems would be helpful.

---

> ### Author Rebuttal · Authors · 2025-07-29
>
> Thank you for your valuable feedback on the readability of our paper and the novelty of our lower bounds. We agree that providing more intuition makes the core ideas more accessible, and we will expand the proof sketches in a revised version of the paper.
>
> ## Regarding the top $k$ components
> We believe our techniques can be extended to the top $k$ components via a deflation-based approaches (see [9,10]). Currently, our algorithm assumes the gapped setting where $\lambda_1 > \lambda_2$. Extending it to the top $k$ components may require the assumption and knowledge of all eigengaps $\lambda_1 - \lambda_2, \lambda_2 - \lambda_3, \dots, \lambda_{k}-\lambda_{k+1}$.
>
> We discuss what we believe is a barrier to extending our analysis to k-PCA with the deflation-based method, as requested by the reviewer, below. However, we want to point out that we believe that k-PCA methods that maintain $k$ vectors and perform QR decomposition at every step (see e.g. [5, 14]) would provide sharper bounds with less stringent assumptions. This is part of ongoing work.
>
> ### **Deflation based methods**
> A standard deflation approach splits the data stream into $k$ chunks. For each chunk $C_i$, construct the projection matrix $P_{i-1}$ that removes the first $i-1$ estimated eigenvectors, and then estimate the leading eigenvector of $P_{i-1} \left(\sum_{j\in C_i} X_j X_j^{\top}\right) P_{i-1}$. Because Oja’s updates use only matrix–vector products, the method remains fully streaming with $O(kd)$ memory. For quantized streaming PCA, one must also consider the accumulation of quantization errors from each eigenvector estimate. We expect that the martingale-based arguments should hold since the new datapoints become $P_{i-1}X_j$ for some fixed matrix $P_{i-1}$ (independent of $X_j$ in chunk $C_i$). However, the main barrier appears from the fact that one needs to make sure that the eigengap $\lambda_i-\lambda_{i+1}$ is larger than the errors accumulated up until now, which is $O(i\Delta)$ where $\Delta$ is the maximum $\sin$ error over the $i$ chunks.
>
> ## Regarding assumption 1
> Assumption 1 states standard moment bounds used to analyze PCA in the stochastic setting. These assumptions are also used in citations [1-7] to derive near-optimal sample complexity bounds for Oja’s rule. We note that while the current analysis assumes a bounded set, this is for ease of analysis, and it can be generalized to unbounded subgaussian data (see [11,12,13]).
>
> For ease of exposition, we will provide an easy argument that changes the algorithm by considering "truncated" datapoints $Y_i := X_i 1 (||X||_{i}^{2} \leq \alpha_n)$, where the truncation parameter $\alpha_n$  is set to be $c\log(n)$. Thus we replace any datapoint whose squared norm exceeds the truncation parameter by zero. Standard truncation arguments can then show that the sin-squared error of the output of this algorithm w.r.t the original principal eigenvector has the same theoretical guarantee.
>
> Equation (1) in the Section titled “Stochastic PCA” in Accelerated stochastic power iteration (citation [8]) states their assumptions; **in fact, [8] uses a stronger assumption than ours.** The matrix $A$ in their paper is the covariance matrix $\Sigma$, the matrices $\tilde{A}_t = X_t X_t^{\top}$, and the almost sure bound $r = \mathcal{M}$. One difference is that the [8] use the bound $E[\left\lVert XX^{\top} - \Sigma \right\rVert ^2] \le \sigma^2$, while we use the weaker assumption $\left\lVert E[ (XX^{\top} - \Sigma)^2] \right\rVert \le \mathcal{V}$; note that $\left\lVert E[ (XX^{\top} - \Sigma)^2] \right\rVert \le E[\left\lVert XX^{\top} - \Sigma \right\rVert ^2]$ by Jensen’s inequality.
>
> ### **References**
> [1] Moritz Hardt and Eric Price. The noisy power method: a meta algorithm with applications. NeurIPS 2014.
>
> [2] Christopher De Sa, Christopher Re, and Kunle Olukotun. Global convergence of stochastic gradient descent for some non-convex matrix problems. ICML 2015.
>
> [3] Ohad Shamir. Convergence of stochastic gradient descent for PCA. ICML 2016.
>
> [4] Ohad Shamir. Fast stochastic algorithms for SVD and PCA: convergence properties and convexity. ICML 2016.
>
> [5] Zeyuan Allen-Zhu and Yuanzhi Li. First efficient convergence for streaming k-PCA: a global gap-free and near-optimal rate. FOCS, 2017.
>
> [6] Prateek Jain, Chi Jin, Sham M. Kakade, Praneeth Netrapalli, and Aaron Sidford. Streaming PCA: Matching matrix Bernstein and near-optimal finite sample guarantees for Oja’s algorithm. COLT, 2016.
>
> [7] Maria-Florina Balcan, Simon Shaolei Du, Yining Wang, and Adams Wei Yu. An improved gap-dependency analysis of the noisy power method. COLT, 2016
>
> [8] Xu, P., He, B., De Sa, C., Mitliagkas, I., & Re, C. (2018, March). Accelerated stochastic power iteration. In International Conference on Artificial Intelligence and Statistics (pp. 58-67). PMLR.
>
> [9] Jambulapati, Arun, et al. "Black-box k-to-1-PCA reductions: Theory and applications." COLT 2024
>
> [10] Mackey, Lester. "Deflation methods for sparse PCA." Advances in neural information processing systems 21 (2008).
>
> [11] Lunde, Robert, et al "Bootstrapping the error of Oja's algorithm." NeurIPS 2022.
>
> [12] Kumar, S., & Sarkar, P. (2024). Oja's Algorithm for Streaming Sparse PCA. arXiv preprint arXiv:2402.07240.
>
> [13] Liang, Xin. "On the optimality of the Oja’s algorithm for online PCA." Statistics and Computing 33.3 (2023): 62.
>
> [14] De Huang, Jonathan Niles-Weed, and Rachel Ward. Streaming k-PCA: Efficient guarantees for Oja’s algorithm, beyond rank-one updates. COLT, 2021.

---

> > ### Comment · Reviewer_4RCk · 2025-08-04
> > **Thanks for the response**
> >
> > Thanks to the authors for their thorough response, especially regarding Assumption 1. I shall maintain my score.

---

### Decision · Program_Chairs · 2025-09-17

**Decision:**

Accept (poster)

**Comment:**

This work studies quantized Oja's algorithm for streaming PCA showing upper and matching lower bounds for linear quantization and logarithmic quantization.

All reviewers agree that the theoretical contributions of the paper are good. There were some concerns raised about the need to know the spectral gap for the algorithm. This has been somewhat addressed by the authors by pointing out that knowing the spectral gap is somewhat of a standard assumption in previous streaming PCA works. Further, some of the reviewers have expressed concerns about the quality of the empirical evaluation. The authors provide an extended empirical evaluation in the rebuttal which has addressed this concern. Several reviewers had issues with the presentation of the paper. I recommend that the authors include the extra experimental evaluation and implement the suggestions on improving the presentation of the paper.